# Centrosomal actin pool levels regulated by localized PKA set the threshold for T cell polarization

Morgane Simao [1,2], Fabienne Régnier[1,2] & Clotilde Randriamampita [1✉]

## Abstract

**T lymphocyte migration triggered by chemokine stimulation is preceded by cell polarization. The acquisition of this asymmetry requires a profound cell rearrangement, particularly of the cytoskeleton. The mechanism by which a uniform signal triggered by chemokine receptors rapidly leads to this asymmetry is largely elusive. Using cell imaging, we emphasize that the centrosome dictates the position of the polarization axis in T lymphocytes. Mechanistically, we highlight that the T cell shape is controlled by the amount of actin filaments surrounding the centrosome. In resting conditions as well as after chemokine stimulation, the activity of a specific pool of protein kinase A regulates this cytoskeleton compartment. Once the centrosomal actin is reduced below a certain threshold, the symmetry breaking is catalyzed. This study points to a critical protein kinase A signaling pathway in the establishment of the immune response.**

**Keywords** Lymphocyte; Polarization; Centrosome; Actin; Protein Kinase A
**Subject Categories** Cell Adhesion, Polarity & Cytoskeleton; Immunology

## Introduction

T lymphocyte migration constitutes a key step of the immune response, considering that it allows the cells to explore primary lymphoid organs as well as tissue. It is conditioned by physical and chemical microenvironment (Krummel et al, 2016) and is thus a result of a combination of 3D confinement, durotaxis, topotaxis, chemotaxis, haptotaxis, etc. which may deliver opposite guidance cues (Shellard and Mayor, 2020; Kameritsch and Renkawitz, 2020). Chemical signals are in particular due to bound or soluble chemokines (CXCL12, CCL19, and CCL21). T cell stimulation by chemokine triggers a major rearrangement of the cell compartments (organelles, cytoskeleton, …) (Pineau et al, 2023) that leads to an asymmetrical cellular organization with two main structures: the lamellipodium at the front and the uropod at the back. This cell polarization is an absolute prerequisite for the cell to migrate and requires important remodeling of the cytoskeleton. In particular, chemokine triggers an increase of actin polymerization, which

progressively accumulates at the cell front, conferring a highly dynamic lamellipodium (Real et al, 2007). The uropod at the cell back is surrounded by an actomyosin network whose contraction facilitates T cell migration. The establishment of this polarization axis takes place within a few minutes after chemokine stimulation. Chemokine receptor activation has been reported to trigger several signaling pathways, such as calcium signaling, ERK, PI3 kinase activation, small GTPases activation, or cAMP modulation (Kehrl, 2006; Thelen and Stein, 2008). Although these different elements seem to be important for T cell polarization and/or migration (Niggli, 2014), their importance in the initiation of polarization, i.e., in early (seconds-minutes) steps leading to the establishment of the asymmetric distribution of cellular components and polarity regulators, is still unclear (Mastrogiovanni et al, 2022).

T cell polarization appears in two important immune functions: upon immune synapse formation and after chemokine stimulation. In the first case, stimulation takes place at a specific location imposed by the point of contact with the antigen-presenting cell, where a signaling pole is formed locally. The position of the polarization axis is then driven by the site of stimulation. Conversely, upon chemokine addition, the initial stimulation might be homogeneous or heterogeneous (gradient of soluble or bound chemokine). However, in both cases, it rapidly triggers a symmetry breaking which allows the cell to polarize in one clear direction after a couple of minutes. One could therefore wonder whether, after a uniform stimulation, the orientation of this axis takes place randomly, or whether it is imposed by an intrinsic cellular factor. In other words, is it possible in that case, to predict in which direction a cell will polarize after uniform chemokine stimulation?

Among the intrinsic cellular factors which could condition the position of the polarization axis, various hypothesis can be considered. An uneven distribution of chemokine receptors at the cell surface might lead to the formation of several local signaling poles. The pole with the highest receptor density might force the development of the polarization axis at a specific place. A heterogeneous distribution of polymerized actin or factors promoting this process, could also favor the formation of the lamellipodium in a specific position after chemokine stimulation. Finally, the centrosome, a single-copy organelle, is now clearly considered not only as a microtubule-organizing center (MTOC) but also as a signaling platform (Langlois-Lemay and D'Amours, 2022) which may consequently transform a uniform external signal into a local response. As a non-membrane organelle, the centrosome is surrounded by a pericentriolar material linked to

[1]Université Paris Cité, CNRS, Inserm, Institut Cochin, F-75014 Paris, France. [2]These authors contributed equally: Morgane Simao, Fabienne Régnier.
✉E-mail: c.randriamampita@inserm.fr

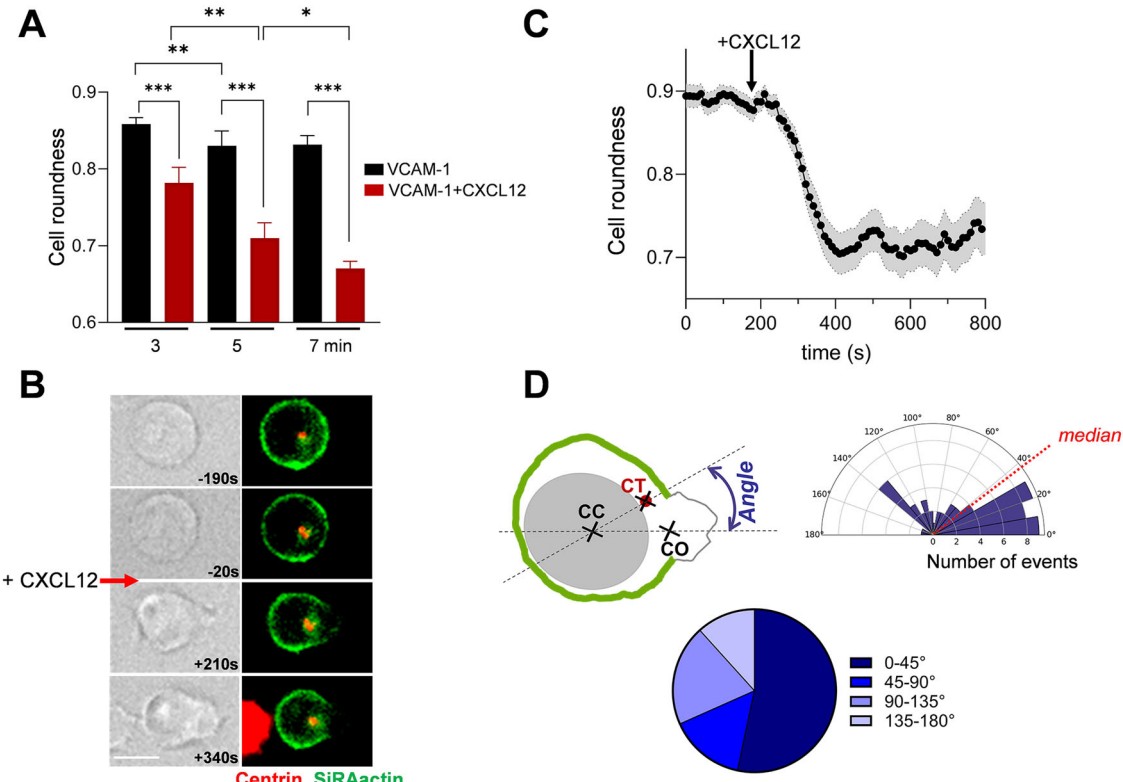

**Figure 1. Chemokine stimulation triggers actin remodeling along an axis conditioned by MTOC position.**

(A) Cell roundness progressively decreases in CEM T cells deposited for different periods of time, on coverslips coated with VCAM-1 + CXCL12. A limited deformation takes place with VCAM-1 alone. Mean ± SE of four independent experiments, 448–1722 cells/condition/experiment. Statistics: Nonparametric ANOVA test for multiple comparisons. ***$p < 0.001$, **$p \leq 0.006$, *$p = 0.048$. (B) Acute addition of CXCL12 triggers rapid cell deformation (Mean ± SE of 48 individual cells). (C) Example of actin cortex opening after CXCL12 stimulation (red arrow) in a CEM T cell transfected with GFP-Centrin1 (red) and whose actin cortex has been labeled with SiRActin (green). Scale bar = 10 μm. (D) As depicted in the scheme (top left), the angle formed by the two lines [center of the cell (CC) - center of the opening of the cortical actin cage (CO)] and [center of the cell—centrosome (CT)] has been quantified in 60 cells. The distribution of the angles measured in similar experimental conditions as (C) is shown in the rose (up right) and pie plots (bottom). Source data are available online for this figure.

the cytoskeleton. A large number of proteins are indeed associated with this structure, including protein kinases and phosphatases (Schatten, 2008). It is involved in several cellular processes (Elric and Etienne-Manneville, 2014), in particular in the positioning of subcellular structures upon immune synapse formation (Kloc et al, 2014).

Recently, the centrosome has been reported to not only be a microtubule, but also an actin organizing center (Farina et al, 2015). In T and B lymphocytes, a decrease in the pool of polymerized actin that surrounds the centrosome has been reported upon immune synapse formation (Obino et al, 2016; Cassioli et al, 2021; Ibanez-Vega et al, 2019; Bello-Gamboa et al, 2020). Furthermore, a recent study demonstrates that this pool of actin could constitute a size-dependent diffusion barrier around the centrosome (Cheng et al, 2022), suggesting that any event leading to actin depolymerization at this point, could induce a modification of the pericentriolar composition. Interestingly, this is what is observed in B lymphocytes, where the nature of the proteins associated with the centrosome is modified upon immune synapse formation (Farina et al, 2015).

The aim of this study is to understand whether, after uniform chemokine stimulation, the symmetry break that drives the position

of the polarity axis is established in a particular direction with respect to the intracellular compartments or whether this is completely random. Using a combination of imaging and pharmacological perturbations, we identify, in the CEM T cell line as well as in human primary T cells, the early signaling pathways which control this event and show that a specific pool of PKA is able to control T cell polarization by regulating the level of actin around the centrosome.

## Results

### CXCL12-stimulated T cells polarize along an axis correlated with the centrosome position

T cell stimulation by chemokine triggers a very rapid cell deformation that precedes polarization itself as previously reported (Real et al, 2007). This deformation can be quantified by the cell roundness parameter, which allows us to calculate how close to a circle the cell is. While the cell roundness is equal to 1 for a perfectly round cell, it decreases as long as the cell deforms. As shown in Fig. 1A, unstimulated CEM T cells deposited on VCAM-

1-coated coverslips present a resting roundness of around 0.85 independently of the duration of adhesion. Conversely, when the coverslips are coated with VCAM-1 plus CXCL12, the cells progressively deform. In this case, the cell stimulation can take place asynchronously depending on the time a given cell enters into contact with the CXCL12-coated coverslip. The kinetics of cell deformation can therefore be more precisely assessed by dynamic cell imaging when cells deposited on VCAM-1 coated coverslips, are stimulated at the same time by the external addition of CXCL12. As depicted in Fig. 1B, in this configuration, we observed that CEM deformation starts rapidly after stimulation with a mean delay between cell stimulation and initiation of cell deformation of $112 \pm 10$ s ($n = 48$ cells). As we have previously shown (Simao et al, 2021), cell polarization corresponds to a major actin cytoskeleton reorganization with an accumulation of newly polymerized actin at the front (lamellipodium), while the stable actin network becomes restricted to the cell back (uropod). This actin reorganization settles the polarization axis. We next considered whether the acquisition of this polarization axis takes place in any direction or along a predetermined position.

The centrosome, which we visualized by transfecting the cells with a GFP-centrin1 construct, can be used as a reference for determining the position of the polarization axis. We followed the establishment of this axis by determining the point where the cortical actin cage (labeled with SiRActin as previously shown (Simao et al, 2021)) opens, allowing the formation of the future lamellipodium. In the example presented in Fig. 1C, it clearly appears that this event takes place in the vicinity of the centrosome. As depicted on the scheme presented in Fig. 1D, we quantified this observation by measuring the angle formed between the two lines [Center of the cell - Center of the opening of the cortical actin cage] and [Center of the cell - Centrosome]. An angle of 0° would mean that the lamellipodium forms exactly in front of the centrosome. Angle distribution presents a mean value of $59.7° \pm 6.6$ ($n = 60$ cells from 11 independent experiments) with a median of 37.5° (Fig. 1C), meaning that the opening of the cortical actin cage preferentially takes place close to the centrosome.

We then investigated the distribution of the chemokine receptor CXCR4 on unstimulated CEM T cells to determine whether they display an accumulation near the centrosome. Such a heterogeneous distribution could indeed trigger a local signaling pole and therefore impose the position of the polarization axis at this place. However, as observed in the different cells presented in Fig. EV1A left, CXCR4 labeling, although not uniform, does not present a systematic enrichment at the centrosome pole. In CXCL12-activated cells, the receptors clearly accumulate at the cell front (Fig. EV1A right). CXCR4 distribution can therefore not explain the preferential position of the polarization axis we observe.

## Protein Kinase A activation triggered by CXCL12 stimulation is essential for cell polarization

We next focused on the signaling pathways triggered during early polarity establishment, in particular the cAMP, PKA, and calcium (Ca) pathways. By using the FRET-based cAMP biosensor TEpacVV (Klarenbeek et al, 2015) and dynamic cell imaging, we analyzed the variations of cAMP after CXCL12 addition. Surprisingly, despite a supposed coupling of CXCR4 to Gi proteins, we observed a very rapid rise in cAMP that reaches its maximum

within a few seconds (Fig. 2A). Because, chemokine stimulation is known to induce a Ca increase through Gβγ-activated phospholipase C (Thelen and Stein, 2008), we wondered whether Ca could be responsible for the cAMP increase through a well-described Ca-cAMP crosstalk (Arige and Yule, 2022). To answer this question, we simultaneously measured Ca and cAMP variations in individual CEM T cells. As shown on the averaged traces presented in Fig. EV1B, cAMP increases more slowly than Ca after CXCL12 addition. These results suggest that the CXCL12-induced Ca rise preceded the cAMP increase. In order to directly test the causal link between the two events, the sensitivity of the cAMP increase to the Ca chelator BAPTA was investigated. As shown in Fig. EV1C, the CXCL12-induced cAMP rise is significantly reduced when Ca is buffered, suggesting that the elevation of cAMP is a consequence of the previous Ca increase.

We next investigated the consequences of the cAMP rise by measuring the variations of PKA activity using the PKA biosensor exRaiAKAR2 (Zhang et al, 2020). As shown in Fig. 2B, an increase in PKA activity is rapidly measured after CXCL12 addition. Analysis of cAMP and PKA responses in a large number of cells, reveals that cAMP levels start to increase $17.63 \pm 0.63$ s ($n = 139$ cells) after CXCL12 addition while PKA activity rises significantly later ($23.33 \pm 0.75$ s, $n = 201$ cells) for (Fig. 2C), suggesting not surprisingly, that cAMP increase precedes PKA activation.

Using dynamic cell imaging, we were able to establish that cAMP increase as well as PKA activation take place before the cells start to deform, as shown in the examples presented in Fig. 2C. This observation can be quantified in single cells by measuring PKA activity simultaneously with roundness evolution over time. It clearly appears that while PKA is rapidly activated, cells deform more slowly and with a certain delay, as shown in the example presented in Fig. 2E. This delay is $72.3 \pm 9.0$ s ($n = 30$ cells). These results demonstrate that CXCL12 triggers first a cAMP rise, followed by an activation of PKA, which itself precedes CEM T cell deformation.

The causal link between these events was directly investigated by testing whether PKA activation was necessary for the establishment of cell polarization. For this purpose, we used H89 as an inhibitor of PKA. This drug is a competitive inhibitor of ATP on kinases catalytic site, in particular that of PKA, although a limited number of other kinases can also be affected at the low H89 concentration we used (Davies et al, 2000; Murray, 2008). As shown in Fig. 2F and on the example presented in Fig. EV2A, H89 prevents CXCL12-induced cell deformation. Interestingly, the roundness of unstimulated cells is also increased with this drug, suggesting that resting PKA activity also controls the shape of unstimulated cells. To complete this result, we performed some experiments on human primary blood T lymphocytes (PBT). In the same way as observed with the CEM T cell line, the deformation of PBT induced by the chemokine CXCL12 is also abolished by H89 pretreatment (Fig. EV3A,B). However, one can note that with these primary T cells, H89 only slightly (and insignificantly) increases the resting rounding. It is probably due to the fact that resting PBT are already very round. They indeed display a resting roundness statistically superior to that of CEM ($0.90 \pm 0.01$ mean of four experiments for PBT vs $0.83 \pm 0.01$ mean of six experiments for CEM).

To analyze the consequences on T cell behavior of H89-induced inhibition of deformation, we quantified the migration by Transwell assay. As displayed in Fig. EV2B, after 2 h, CXCL12 triggers the

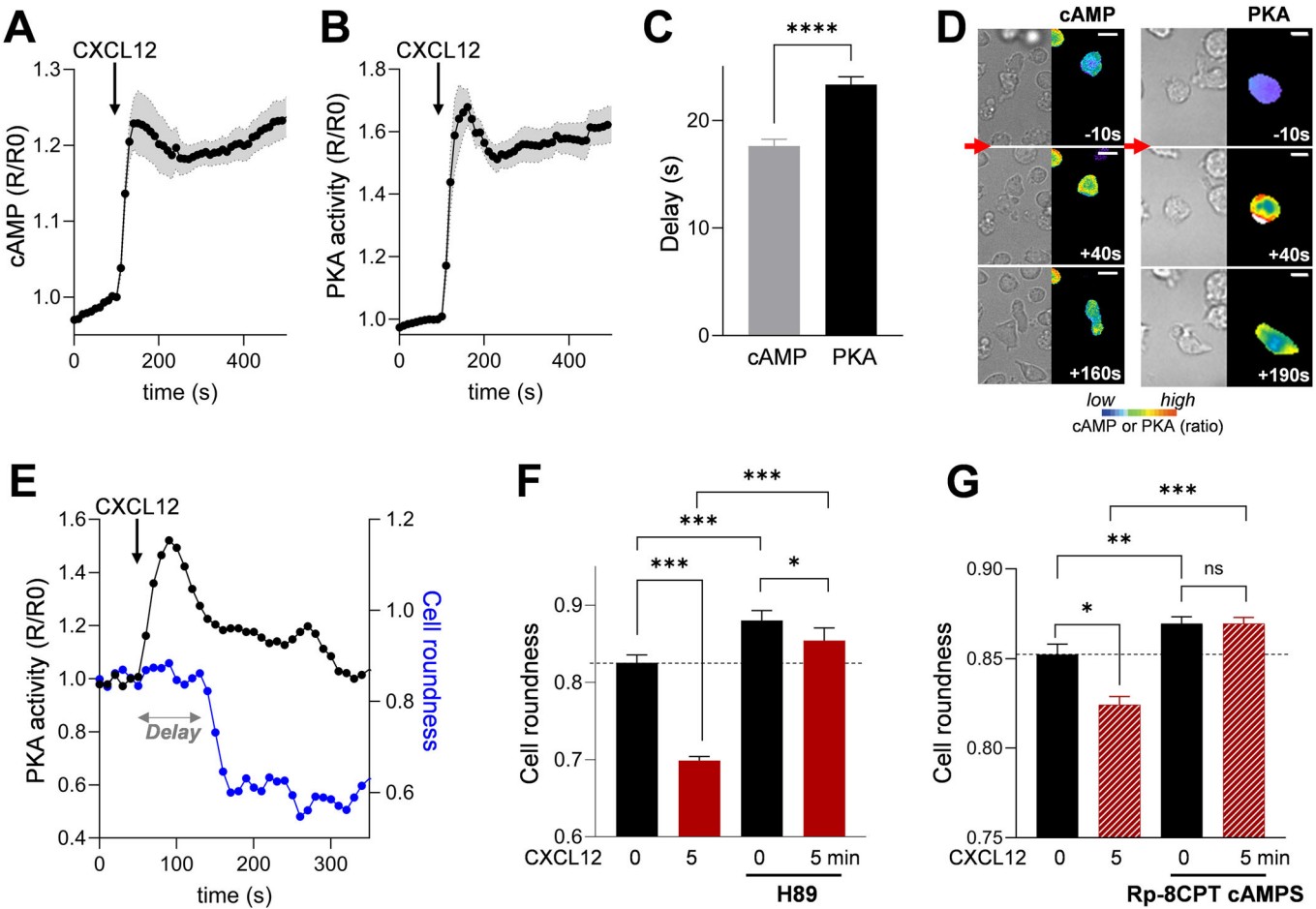

**Figure 2. Chemokine stimulation triggers a rapid increase in cAMP and activation of PKA necessary for cell deformation.**

(A) CXCL12 addition triggers a rapid rise in cAMP in CEM T cells transfected with TEpacVV biosensor. Mean ± SE of six independent experiments, 6–59 cells/experiment. (B) CXCL12 addition triggers a rapid rise in PKA activity in CEM T cells transfected with ExRaiAKAR2 biosensor. Mean ± SE of six independent experiments, 11–46 cells/experiment. (C) Quantification of the delay between CXCL12 addition and cAMP or PKA activity increases (first point above the baseline). Mean ± SE of 139 cells (cAMP) and 201 cells (PKA). Statistics: Unpaired *t*-test ****$p < 0.0001$. (D) Examples of CEM T cells transfected with TEpacVV (left) or ExRaiAKAR2 (right) and stimulated with CXCL12. cAMP (left) or PKA activity (right) are low in resting cells, but rapidly increases after CXCL12 addition (red arrow). Cell deformation starts thereafter. cAMP and PKA activity are coded in false color. Time 0: CXCL12 addition. Scale bar = 10 μm. (E) Example of a simultaneous measurement of PKA activity (black line) and cell roundness (blue line) in a CEM T cell transfected with exRaiAKAR2. The delay between the increase of PKA activity and the beginning of cell roundness decrease is indicated by the gray arrow. (G) Pretreatment with H89 (F) or Rp-8-CPT-cAMPS (G) of CEM T cells deposited on VCAM-1 or VCAM-1 + CXCL12-coated coverslips for 5 min, induces an increase of basal cell roundness and a reduction of the chemokine-induced cell deformation. CXCL12 1 μg/ml in (F) and 0.3 μg/ml (G). Mean ± SE of 6 independent experiments, 264–1475 cells/condition/experiment (F). Mean ± SE of six independent experiments, 479–1283 cells/condition/experiment (G). Statistics: Nonparametric ANOVA test for multiple comparisons. (F): ***$p < 0.001$, *$p = 0.011$. (G): ***$p < 0.001$, **$p = 0.004$, *$p = 0.024$. Source data are available online for this figure.

migration of 31 ± 7% ($n = 3$) of CEM T cells through 5 μm pores. In the presence of H89, a 60 ± 1% inhibition ($n = 3$) of cell migration was observed, demonstrating that the inhibition of CXCL12-induced cell deformation results in reduced cell migration.

Finally, to confirm the involvement of PKA in this process, we used another family of PKA inhibitors: the cAMP analog Rp-8-CPT-cAMPS, a competitive inhibitor of PKA which binds to the cAMP binding site of the regulatory subunit. Unlike H89, Rp-8-CPT-cAMPS displays a high specific inhibition toward PKA (Gjertsen et al, 1995). As shown in Fig. EV4A, a similar increase in resting roundness is observed when the cells are incubated with this inhibitor. However, this inhibitor is not efficient enough to block the deformation induced by chemokine (Fig. EV4A) unless the strength of the stimulation is

reduced by a factor of 3 (Fig. 2G). It suggests that, under our conditions, this inhibitor is only able to counteract mild activation of PKA. Nevertheless, this result confirms the central role of PKA in CXCL12-induced cell deformation.

## Upon CXCL12 stimulation, centrosomal F-actin decreases through a PKA-dependent mechanism

We then wondered what the link might be between the centrosome associated with the position of the polarization axis (Fig. 1D) and the protein kinase A which is required for deformation (Fig. 2). During the last few years, several papers have reported that, upon immune synapse formation (another example of T cell

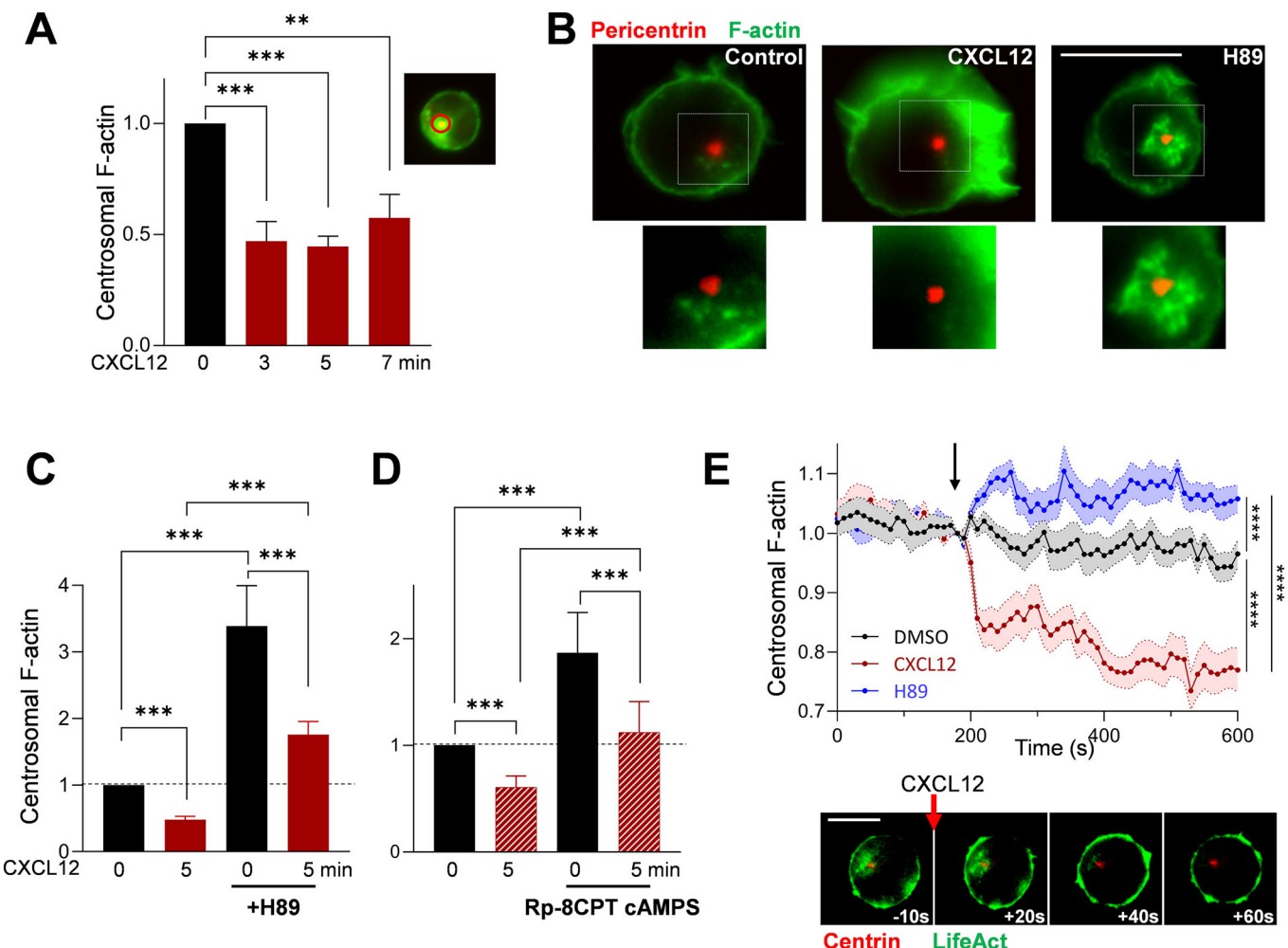

**Figure 3. Chemokine stimulation triggers PKA-induced decrease of polymerized actin at the centrosome.**

(A) Polymerized actin around the centrosome measured in a 3-μm diameter area (inset) decreases progressively in CEM T cells deposited on VCAM-1 or VCAM-1 + CXCL12 for the indicated duration. For each experiment, values have been normalized relative to the mean intensity of centrosomal F-actin measured in unstimulated cells. Mean ± SE of 8–10 independent experiments, 21–159 cells/condition/experiment. Statistics: Nonparametric ANOVA test for multiple comparisons. ***$p < 0.001$, **$p = 0.006$. (B) Examples of F-Actin (labeled with phalloidin) distribution around the centrosome (detected by anti-pericentrin antibodies) in untreated cells (control), stimulated with CXCL12, or simply pretreated with H89. Scale bar = 10 μm. Zooms of the indicated regions are shown below. (C, D) CEM T cells preincubated with H89 (C) or Rp-8-CPT-cAMPS (D) display a higher level of centrosomal actin in resting conditions compared to non-stimulated cells. After 5 min stimulation with CXCL12 (100 ng/ml (C) or 25 ng/ml (D)), this content is reduced but still remains above that of resting cells (dotted line). For each experiment, values have been normalized relative to the mean intensity of centrosomal F-actin measured in untreated and unstimulated cells. Mean ± SE of five independent experiments, 22–165 cells/condition/experiment (C). Mean ± SE of four independent experiments, 36–94 cells/condition/experiment (D). Statistics: Nonparametric ANOVA test for multiple comparisons. ***$p \leq 0.001$. (E) Dynamic evolution of centrosomal actin in CEM T cells transfected with LifeAct-mCherry and GFP-centrin1 after addition (arrow) of DMSO (black trace), CXCL12 (red trace), or H89 (blue trace). For each cell, intensity values have been normalized relative to the value at the time of stimulation. Mean ± SE of 19–30 cells/condition. Lower panel: Example of centrosomal actin evolution after CXCL12 addition (red arrow) to a CEM T cell transfected with LifeAct-mCherry and GFP-centrin1. Scale bar = 10 μm. Source data are available online for this figure.

polarization), important remodeling takes place around T or B cell centrosomes (Obino et al, 2016; Cassioli et al, 2021; Ibanez-Vega et al, 2019; Bello-Gamboa et al, 2020). More precisely, a decrease in the F-actin content in this region has been reported. We therefore studied this particular actin pool over chemokine-induced polarization.

In order to follow the quantity of F-actin around the centrosome upon CXCL12 stimulation, we measured over time, the intensity of the phalloidin labeling in a 3 μm diameter area centered on the centrosome, whose position is determined by pericentrin labeling

(Fig. 3A, insert). The kinetics highlight that centrosomal F-actin rapidly decreases after CXCL12 addition (Fig. 3A and example in Fig. 3B), reaching a minimum already around 3 min. To investigate whether this event depends on PKA activity, we performed the same kind of experiment but in the presence of the PKA inhibitor H89. In the presence of H89, F-Actin content in the centrosomal zone decreases less after CXCL12 addition, compared to control cells (Fig. 3C). Indeed, even after 5 min stimulation, centrosomal F-actin is still above the level measured in unstimulated control cells (dashed line). Furthermore and interestingly, in these

conditions, the level of centrosomal F-Actin in unstimulated cells is dramatically increased by a factor of $2.29 \pm 0.39$ ($n = 8$ independent experiments) (Fig. 3B,C). Similarly to what is observed with the CEM T cell line, CXCL12 stimulation in PBT triggers a decrease of centrosomal actin, which is abolished under PKA inhibition by H89 (Fig. EV3C,D). Furthermore, as noticed in CEM T cells, the resting level of centrosomal actin is also strongly enhanced by this inhibitor in PBT.

An effect similar to H89 was observed when PKA was inhibited with the specific PKA inhibitor, Rp-8-CPT-cAMPS. An increase in centrosomal actin is observed in unstimulated cells incubated with this inhibitor. After CXCL12 stimulation, the decrease of centrosomal actin level is abolished, but, like for cell roundness, only if the intensity of the stimulation is reduced (Figs. 3D and EV4B).

In order to have better kinetic resolution, by dynamic imaging, we measured the centrosomal F-Actin decay in single CEM T cells expressing the polymerized actin reporter, LifeAct. In this configuration, CXCL12 drives a very rapid F-actin decrease immediately after CXCL12 addition (Fig. 3E). Interestingly, H89 on its own, is able to induce a rapid increase in F-actin, suggesting once again that, even in resting cells, basal PKA activity actively controls the level of F-actin content at the centrosome.

Inspired by the results obtained upon immune synapse formation in B cells (Obino et al, 2016), where the decrease of centrosomal F-actin seems to be driven by a decrease in Arp2/3 content in this area, we measured in individual cells, the Arp2 content together with F-actin in the centrosomal zone in control condition, after CXCL12 stimulation or in the presence of H89 (example in Fig. EV4C). Clearly, Arp2 presents a similar evolution to F-Actin: it is decreased after 5 min of CXCL12 stimulation, while it is increased by H89 treatment (Fig. EV4D).

## Centrosomal F-Actin is controlled by a specific pool of PKA

Altogether, these results show that PKA is able to regulate the pool of centrosomal actin in resting cells, as well as after CXCL12 stimulation. One could then wonder whether the PKA activity responsible for this effect is equally distributed in the cell, or restricted to this specific region. By measuring PKA activity through dynamic cell imaging, we sometimes but not systematically observed some hot spots of activity around the centrosome. The variability of this observation might be due to the low frequency of acquisition (one measurement every 10 s). In cells, PKA activity might be compartmentalized at specific intracellular locations through the family of scaffold proteins AKAP (A-Kinase anchoring protein). One specific AKAP, AKAP9 (also called AKAP450, AKAP350, or CG-NAP) has been reported to accumulate at the centrosome and, interestingly, to be required for T cell migration (El Din El Homasany et al, 2005; Ong et al, 2018). As shown in the example (Fig. 4A), in CEM T cells, AKAP450 is present and colocalizes with pericentrin, as attested by the value of the Pearson coefficient ($0.78 \pm 0.01$ $n = 37$ cells). AKAP450 is a type II AKAP, i.e., it is able to bind RII subunits of PKA. To determine whether this AKAP enables compartmentalization of the PKA activity responsible for centrosomal F-actin regulation, we used Ht31. This peptide is able to specifically counteract the binding of type II PKA to AKAP and thus prevents compartmentalization of this class of PKA, especially those bound to AKAP450. To test the efficacy of

this inhibitor, we compared the distribution of PKA using an antibody directed against the catalytic subunit, in the presence of the inhibitory peptide (Ht31) or of the control peptide (Ht31-P). As displayed in the examples presented in Fig. EV5A, an enrichment of PKA in the centrosomal area was observed in control conditions (Ht31-P), while a more homogeneous labeling within the cytoplasm was detected in the presence of Ht31. This difference was quantified by calculating the ratio between the PKA intensities measured in a 3-µm diameter zone centered at the centrosome with one in the cytoplasm. A ratio >1 indicates an accumulation of PKA at the centrosomal area. As shown in Fig. EV5B, this ratio is a little superior to 1 in control conditions but is statistically reduced in the presence of the Ht31 inhibitor. These results suggest there is some accumulation of PKA around the centrosome, which can be reduced by Ht31. We then tested the consequences of this inhibitor on CXCL12 response. As shown in Fig. 4B left, incubation of the cells with this peptide induces an increase in resting roundness and prevents CXCL12-induced cell deformation, unlike the control peptide (Ht31-P). Furthermore, in the presence of this inhibitor, centrosomal f-actin still remains above the level of control cells even after CXCL12 stimulation (Fig. 4B right). Finally, similarly to what is observed with PKA inhibitors, Ht31 induces a major increase in F-actin content at the centrosome in unstimulated cells, as clearly seen in the examples presented in Fig. EV5A. Altogether, these results suggest that in resting cells, as well as in CXCL12-stimulated ones, a pool of PKA anchored at the centrosome, controls the cell roundness and the quantity of centrosomal F-actin.

## Centrosomal F-actin decrease is accompanied by an increase in centrosomal microtubules

We then wondered what could be the consequences of the decrease of F-actin at the centrosome. Among the different possibilities, one can hypothesize that centrosomal actin decrease permits an increase of microtubule polymerization at this location as previously reported (Inoue et al, 2019). We then quantified the two cytoskeleton networks in the different conditions. As shown in Fig. 4C and in the example presented in Fig. EV5C, while centrosomal actin decreases after CXCL12 stimulation, the level of polymerized microtubules increases. Surprisingly, upon H89 addition, polymerized actin and microtubules both increase, suggesting a direct effect of PKA on microtubule polymerization/depolymerization equilibrium.

## Causal link between centrosomal F-Actin and cell deformation

All the results presented above suggest a link between the level of centrosomal actin and cell shape. We then attempted to investigate the potential causality between the two events. First, we determined the temporality of the two events by following the two parameters in individual cells by dynamic cell imaging. As shown in the example presented in Fig. 4D, after chemokine stimulation, the cell starts to deform (blue line) 70 s after centrosomal F-actin starts to decrease (black line). The average value of the delay between these two events is $56 \pm 12$ s ($n = 26$ individual cells), showing that they are clearly sequential.

WASH is an actin nucleation-promoting factor which is able to activate Arp2/3 through its VCA (Verprolin homology of

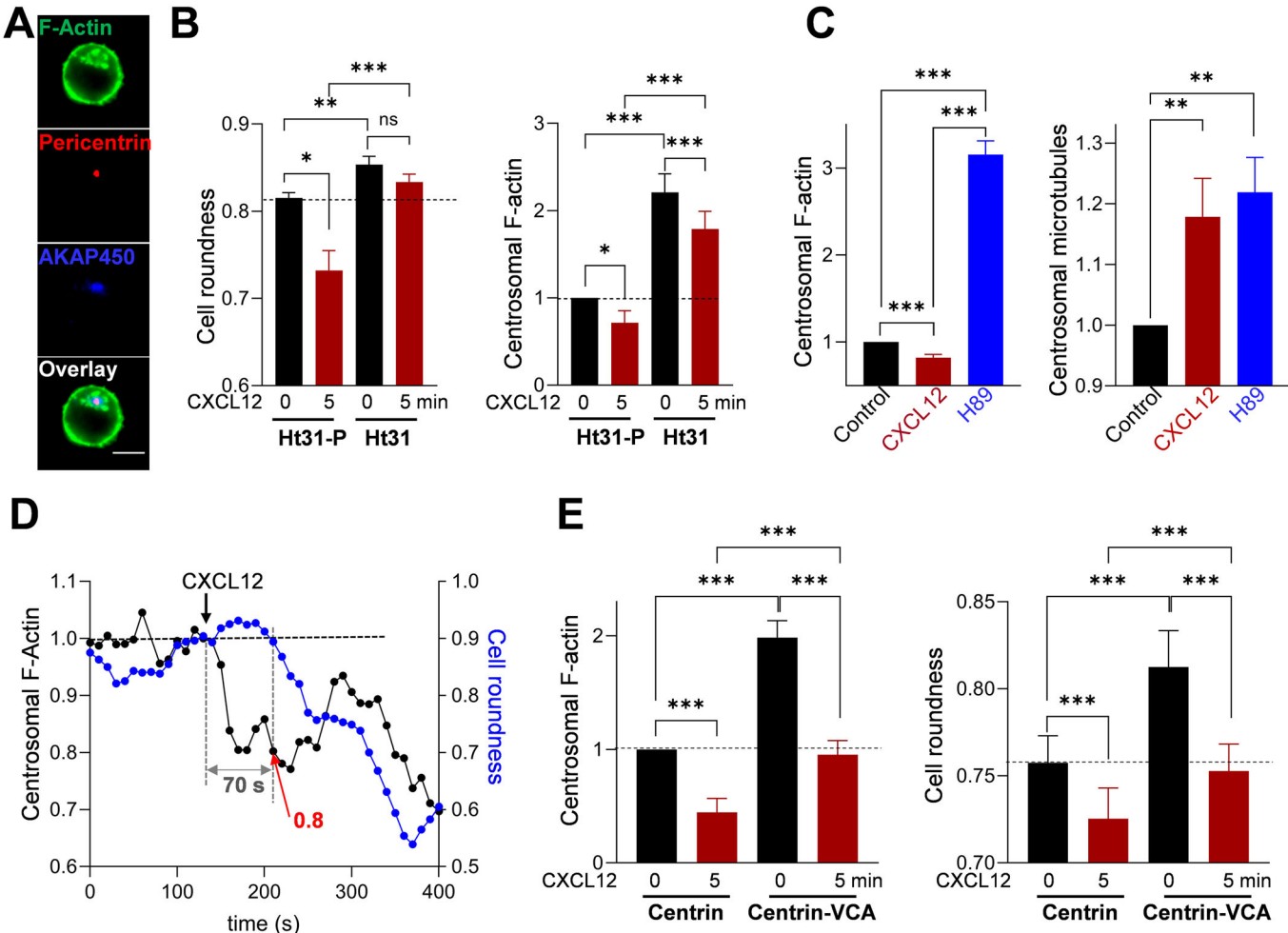

**Figure 4. Direct targeting of centrosomal pools of PKA or actin, alters the response to CXCL12.**

(A) Representative example of distribution of AKAP450, Pericentrin, and F-actin (visualized by specific antibodies and phalloidin, respectively). Scale bar = 10 µm. (B) Left: Pretreatment with Ht31 peptide (AKAP II inhibitor), induces an increase of resting roundness when compared with unstimulated CEM T cells incubated with the control peptide (Ht31-P) (dotted line). After 5 min stimulation with CXCL12, the deformation observed in the presence of the control peptide, is abolished when the AKAP II inhibitor is present. Mean ± SE of five independent experiments, 280–3589 cells/condition/experiment. Right: Pretreatment with the AKAP II inhibitor peptide Ht31, induces an increase of resting centrosomal actin when compared with unstimulated CEM T cells incubated with the control peptide (Ht31-P) (dotted line). After 5 min of stimulation with CXCL12, this content remains above the level measured with the control peptide when the AKAP II inhibitor is present. For each experiment, values have been normalized relative to the mean intensity of centrosomal F-actin measured in unstimulated cells treated with Ht31-P (control peptide). Mean ± SE of five independent experiments. About 30–70 cells/condition/experiment. Statistics: Nonparametric ANOVA test for multiple comparisons. ***$p \leq 0.001$. **$p = 0.04$, *$p \leq 0.023$. (C) The levels of actin (left, phalloidin labeling) and microtubules (right, anti-α-tubulin labeling) were measured concomitantly in CEM T cells in control conditions, after CXCL12 stimulation or H89 pretreatment. For each experiment, values have been normalized relative to the mean intensity of centrosomal F-actin or α-tubulin measured in untreated and unstimulated cells. Mean ± SE of four independent experiments, 29–150 cells/condition/experiment. Statistics: Nonparametric ANOVA test for multiple comparisons. ***$p \leq 0.001$. **$p = 0.01$. (D) Example of simultaneous recording of cell roundness (blue line) and centrosomal actin (black line) in a LifeAct-mCherry-transfected CEM T cell, upon CXCL12 stimulation (black arrow). The gray arrow indicates the time between decreases of centrosomal actin and cell roundness. Red arrow: level of centrosomal actin at which the cell starts to deform. (E) Centrin1-VCA expressing CEM T cells present a higher level of centrosomal actin (left) and an increase in cell roundness (right) compared to Centrin1 expressing cells. In these cells, the effect of CXCL12 is reduced. Left panel: for each experiment, values have been normalized relative to the mean intensity of centrosomal F-actin measured in unstimulated centrin1-transfected CEM T cells. Mean ± SE of 6 independent experiments, 19-69 cells/condition/experiment (left). Mean ± SE of seven independent experiments, 103–437 cells/condition/experiment (right). Statistics: Nonparametric ANOVA test for multiple comparisons. ***$p \leq 0.001$. Source data are available online for this figure.

WH2-connector-acidic) domain. Targeting this domain to the centrosome by expressing the GFP-centrin1-VCA construct permits to artificially trigger an increase of F-Actin at this place as previously reported (Obino et al, 2016). We therefore used this approach to analyze the direct impact of centrosomal F-Actin on cell roundness. In CEM T cells, expression of the centrin1-VCA construct induces, as expected, a clear accumulation of F-Actin at the centrosome in non-

stimulated T lymphocytes, as displayed in the examples (Fig. EV5D) and quantified in Fig. 4E (left). Interestingly, the cells in which centrosomal F-Actin was artificially increased by VCA expression, are less deformed as shown by a higher roundness value (Fig. 4E right and examples in Fig. EV5E). Therefore, in resting cells, increasing the level of F-actin at the centrosome is sufficient to trigger a rounding of the cells. In GFP-centrin1-VCA expressing cells, the CXCL12-

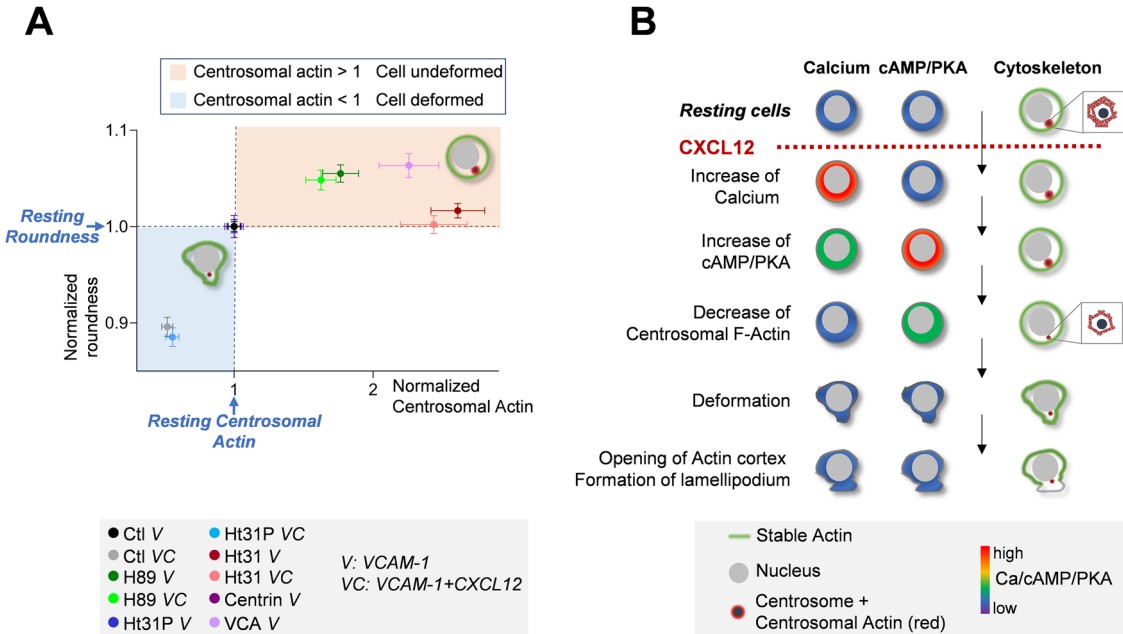

**Figure 5.  Link between PKA, centrosomal actin, and cell roundness.**

(A) Correlation between centrosomal actin and cell shape. For the different conditions and for a given cell, centrosomal actin level and roundness have been normalized to the mean values of the control cells of the same experiment. Each dot represents the average ± SE of 86–359 individual cells. (B) Summary of the sequence of events leading to cell polarization after CXCL12 stimulation. Calcium/cAMP/PKA variations are displayed in parallel to cytoskeleton reorganization. Source data are available online for this figure.

triggered signaling pathways leading to centrosomal actin reduction are still efficient. We then observed that, unlike with PKA inhibitor, CXCL12 is able to decrease centrosomal actin as well as to deform the cells. Of note, the values of both parameters measured in these conditions are not significantly different from those measured in unstimulated control cells.

Altogether, our results show that (1) cell deformation appears after the decrease of centrosomal F-Actin and (2) that any treatment, such as PKA inhibitors (H89 and Rp-8-CPT-cAMPS), AKAP inhibitor Ht31, or centrin1-VCA, which leads to an increase in centrosomal F-Actin, also induces a rise in cell roundness. Furthermore, although CXCL12 is still able, in these different conditions, to decrease centrosomal F-actin, it does so moderately, so that centrosomal F-actin always stays above the level measured in control conditions, and the cells remain less deformed than the resting cells. This observation suggests that a decrease of centrosomal F-actin at least below the resting value, has to be reached for the cells to deform. In order to analyze this threshold, we measured concomitantly at the single cell level, centrosomal actin and cell roundness, and determined, after CXCL12 stimulation, the value of actin at which the cell starts to deform. In the example presented in Fig. 4D, the cell starts to deform (blue line) when the centrosomal actin value is 0.8 (i.e., 20% decrease) (red arrow). By analyzing a series of cells, we determined that centrosomal F-Actin has to be at 85 ± 2% (*n* = 26 individual cells) of the initial value for the cell to quit its resting shape.

The link between the level of actin at the centrosome and cell roundness we observed in our experiments, can be visualized by the phase diagram-like graph presented in Fig. 5A, where the roundness has been plotted relative to the level of centrosomal actin. Each dot represents the mean of many cells for which the two parameters

have been measured. In each case, the values of centrosomal F-actin and roundness have been normalized relative to the mean values measured in unstimulated control cells of the same experiment (i.e., no treatment for control, H89; Ht31-P for Ht31; GFP-centrin1 for GFP-centrin1-VCA). Therefore, control cells have (1,1) as coordinates. Four quadrants can then be distinguished relative to the levels of normalized centrosomal actin or normalized roundness. Clearly, the different values distribute into only two of the four quadrants corresponding to round cells with a centrosomal F-actin value >1 (orange area), or to deformed cells with centrosomal F-actin <1 (blue area). The two other configurations are not observed. This observation suggests that centrosomal F-actin content behaves as a switch (when it reaches 85% of the initial level) that toggles the cells from the round to the deformed state.

All the experiments presented above allow us to propose a sequential series of events leading to T cell deformation: chemokine stimulation triggers a rapid rise in Ca, then in cAMP, followed by PKA activation, which triggers a decrease in polymerized actin around the centrosome. This step is a prerequisite for deformation of the cell, which precedes the opening of the cortical actin cortex near the centrosome to establish the polarity axis. This sequence is summarized in Fig. 5B.

## Discussion

T lymphocyte migration is an essential step to mount an efficient immune response (Krummel et al, 2016). It is essentially under the control of chemokines, which trigger T cell polarization and, thereafter, cell motility. These events require complete cellular

reorganization in order for the cell to acquire an asymmetrical distribution of its components (Mastrogiovanni et al, 2022). This then allows the establishment of a lamellipodium and a uropod, which settle the polarization axis and therefore the direction of migration (Pineau et al, 2023). Even in the absence of a chemokine gradient, cells polarize in response to this external signal. In this study, we wondered, how the position of the polarity axis is controlled in these conditions, and what are the signaling pathways involved. Our results obtained on both CEM T cells and primary human T lymphocytes highlight a crucial role of the centrosome and, more precisely, of a PKA-regulated pool of actin in the centrosomal zone.

## PKA and response to chemokine

Our results clearly identify the cAMP/PKA pathway as playing a key role in T cell response to chemokine. In particular, we emphasize the involvement of a specific pool of PKA, anchored at the centrosome through AKAP450. Consistent with our data, the deletion of the protein has been previously reported to prevent T cell deformation and migration upon chemokine stimulation (Ong et al, 2018). Our conclusions rely on temporal correlation between PKA activation and cell deformation/centrosomal actin reduction, but also on pharmacological inhibitors targeting PKA (H89, Rp-8-CPT-cAMPS, and Ht31). At the H89 dose we used, H89 mainly inhibits PKA, although we cannot rule out it also inhibits other kinases in our cells (Davies et al, 2000; Murray, 2008). However, the H89 effect can be mimicked by two other agents specific to PKA: Ht31 and Rp-8-CPT-cAMPS. The fact that Rp-8-CPT-cAMPS is less efficient than H89 is probably linked to its mechanism of inhibition: it competes with endogenous cAMP so that its inhibition potency might be dependent on the quantity of cAMP produced, i.e., on the strength of chemokine stimulation.

The positive involvement of PKA in T cell polarization may be surprising for two reasons. First of all, our results show, that CXCL12 stimulation is able to trigger a rapid rise in cAMP, followed by PKA activation, despite the fact that chemokine receptors are normally coupled to Gi proteins supposed to inhibit adenylate cyclases. However, we show that, in our case, the cAMP rise is mediated by the chemokine-induced increase of calcium, which precedes the cAMP signal. This result may suggest the involvement of an unidentified calcium-sensitive adenylate cyclase (such as AC1, AC3, AC8, or AC10 (Ostrom et al, 2022; Steegborn, 2014)), although AC7 has been reported to be the most abundant adenylate cyclase in T cells (Duan et al, 2010). Secondly, cAMP/PKA have been considered as an inhibitory pathway for T cell migration. This consideration is based on the fact that a large and sustained increase in cAMP triggered by forskolin, PGE2, or phosphodiesterase inhibitors induces a rounding up of the cells (Valitutti et al, 1993; Oppenheimer-Marks et al, 1994; Layseca-Espinosa et al, 2003; Dong et al, 2006). However, after chemokine addition, the rise in cAMP remains modest and transient. It is therefore likely that different features of cAMP signals (intensity and time course) do not lead to the same effect as is the case for calcium, for which it is clearly established that the shape of the signal conditions the cellular response (see (Berry et al, 2018) for a review). Furthermore, with the combination of two properties of AKAP450, it is possible that this protein is able to restrict the PKA signal at its close vicinity. First, it is able to bind different partners,

including the phosphodiesterase PDE4D3 (Tasken et al, 2001), able to degrade cAMP, and then to reduce the scale of the chemokine-induced cAMP increase. Secondly, the binding of PKA to AKAP450 reduces the activation threshold of the kinase by cAMP (Terrin et al, 2012). This could allow a moderate cAMP signal to trigger PKA activation only around the centrosome where AKAP450 is anchored, and then to prevent the previously described inhibitory effect of this kinase.

## Regulation of centrosomal actin in lymphocyte

Recent studies in T and B lymphocytes demonstrate that, upon immune synapse formation, the pool of actin around the centrosome is decreased (Obino et al, 2016; Cassioli et al, 2021; Ibanez-Vega et al, 2019; Bello-Gamboa et al, 2020). Here for the first time, we show that a similar process takes place in response to chemokine. Using dynamic cell imaging and immunofluorescence, we were able to show that this phenomenon takes place almost immediately after chemokine addition. In the case of antigen recognition, the specific kinetics of this process has not been investigated, and the first time point at which the decrease is reported is 15 min, but it is possible that the decrease starts earlier. The level of actin at the centrosome results from a balance between the polymerization and depolymerization processes at this site, so that a decrease of F-Actin after chemokine stimulation might be due to an inhibition of the polymerization and/or activation of the depolymerization. In the context of the immune synapse, several non-exclusive mechanisms have been suggested to explain this reduction in centrosomal actin (Cassioli and Baldari, 2022). Upon T/B synapse, the recruitment of HS1 at the synapse, triggers the displacement of Arp2/3 from the centrosome to the synapse and therefore induces a reduction of F-Actin at this place (Obino et al, 2016). In our conditions, we observed a similar reduction of Arp2/3 in the centrosomal area (Fig. EV4C,D), although we do not know which mechanism drives it. The involvement of PKCδ and of the proteasome have also been shown upon immune synapse formation (Cassioli et al, 2021; Ibanez-Vega et al, 2019; Bello-Gamboa et al, 2020), but no evidence of the involvement of these pathways exists in the case of chemokine stimulation. On the contrary, our results suggest that a local pool of PKA anchored to the centrosome through an AKAP would control the level of F-actin at this place, not only after chemokine stimulation, but also in resting cells. It is still unclear the mechanism by which PKA exerts this effect and what the target(s) it phosphorylates.

In resting cells, the centrosome presents an off-center positioning which corresponds to its minimum-energy position (Maly, 2011) and is controlled by the actin-myosin network (Jimenez et al, 2021). However, during cellular responses, its position seems to be finely regulated (Kroll and Renkawitz, 2024), especially in T lymphocytes during chemokine-triggered polarization or immune synapse formation, where a relocation of the centrosome is observed. Centrosomal actin seems to contribute to the stabilization of the centrosome (Yamamoto et al, 2022) and, in B lymphocytes, to its anchoring to the nucleus through the LINC complex (Obino et al, 2016). One could then imagine that the decrease of centrosomal actin after chemokine stimulation, which we observe in our study, allows the reduction of centrosome anchoring to the nucleus, a necessary step for the relocation of the centrosome behind the nucleus during migration. The front

position of the nucleus constitutes indeed a specific characteristic of ameboid migration, permitting the cell to gauge the environment and to perform optimal migration (Kroll et al, 2023; Renkawitz et al, 2019). Corroborating this hypothesis, with a siRNA screen, Obino et al recently identified, which proteins among those differentially associated with the centrosome upon B cell stimulation, are important for centrosome movement. Type II regulatory subunit beta and catalytic subunit beta of PKA appear in this list (Obino et al, 2023).

Our study highlights a central role of PKA-regulated centrosomal actin upon uniform chemokine stimulation. However, one can wonder what would happen in the case of a chemokine gradient, as frequently observed in vivo. Would the cells polarize along the gradient independently of the centrosome position? Would the centrosome, whatever its position relative to the gradient, impose the polarization axis? Or, finally, as a mix of the two previous possibilities, would the cell rotate in order to position the centrosome according to the gradient?

## Centrosomal actin, cell deformation, and lamellipodium formation

Besides the potential role of centrosomal actin in controlling the centrosome mobility, our results suggest that it also regulates the shape of the cells in resting conditions, as well as after chemokine stimulation. What is the link between these two events? Several non-exclusive hypotheses can be suggested. Firstly, it has been shown recently that actin (rather than microtubules) around the centrosome can be a cage that limits the exchange between the cytoplasm and the pericentriolar region (Cheng et al, 2022). A decrease of actin in this area would then weaken this barrier and potentially lead to modification of the centrosomal composition. Interestingly, that is what is observed by Obino et al, who showed, in parallel to immune synapse formation, a decrease in centrosomal actin as well as a modification of the protein content associated with the centrosome: some leave this area, while some are recruited (Obino et al, 2016). A similar process could probably take place in our case, after stimulation by chemokine. Because some of these proteins belong to signaling pathways, this local actin modification will influence local signaling events which may be important for centrosome relocation and for the cellular response. Among these proteins, as mentioned above, Arp2/3 leaves the centrosome upon immune synapse formation, as well as after chemokine stimulation (Fig. EV4C,D). This major actin branching protein can be considered a fundamental brick for actin network formation at the lamellipodium, so that triggering its release is crucial. On the other hand, the disappearance of the actin cage around the centrosome seems to also allow the recruitment of new proteins at this site. This is particularly the case of Dock2, an exchange factor of the small GTPase Rac, a crucial element for actin polymerization in T cells. It is then possible that the local activation of Rac through Dock2 triggers actin polymerization, and the subsequent formation of the lamellipodium in this area. Finally, the decrease of the centrosomal actin pool can have some consequences on the microtubule network, which controls cell shape and contributes to cell directionality, although it is dispensable for T cell migration (Kopf and Kiermaier, 2021). Indeed, actin and microtubule networks at the centrosome have been reported to be dynamically connected so that actin level tunes microtubules and vice versa (Inoue et al, 2019). This might result from the fact that centrosomal

F-Actin could constitute a physical barrier preventing microtubule elongation (Cheng et al, 2022). The local decrease of centrosomal actin could therefore be essential for the extension of the microtubule network and the subsequent centrosome repositioning (Yamamoto et al, 2022). In line with these results, we have observed in our model that CXCL12-induced decrease of F-actin at the centrosome is accompanied by an increase of microtubules in this area and can then be essential for T cell polarization. However, in an unexpected way, the PKA inhibitor H89 triggers increases of both polymerized actin and microtubules. This suggests that, besides the supposed balance between the two cytoskeleton networks, PKA might directly promote microtubule depolymerization (or inhibit their polymerization), for instance, through microtubule-associated proteins whose activity can be affected by phosphorylation (Ramkumar et al, 2017).

Our results highlight that a $15 \pm 2\%$ reduction of centrosomal actin switches the cells from a round state to a deformed one (summary in Fig. 5A). The biological meaning of this threshold is unclear, but it could correspond to the reduction of the size of the actin mesh. Indeed, Cheng et al highlight that centrosomal actin behaves as a protein sieve, and that the reduction of its amount allows the centrosome accessibility of proteins with a higher Stokes radius (Cheng et al, 2022). One could then imagine that a 15% reduction of centrosomal actin is the necessary decrease for recruiting (or losing) the specific proteins, mentioned above, which are necessary for the T cell morphological change.

Altogether, our results highlight a central role of PKA in the establishment of the polarization axis upon chemokine stimulation. These results establish a new example where transient activation of PKA promotes an efficient T cell response (Simao et al, 2021; Conche et al, 2009), contrary to large and sustained activation which triggers some immunoinhibitory effects (Postler, 2021).

## Methods

**Reagents and tools table**

| Reagent/resource | Reference or source | Identifier or catalog number |
|---|---|---|
| **Experimental models** | | |
| CCRF-CEM | Megrelis et al (2018) Frontiers in Immunology | Jérôme Delon |
| **Recombinant DNA** | | |
| LifeAct-mCherry | Molla-Herman A et al (2010) (Journal of Cell Science) | Dr A. Benmerah (Institut Imagine, France) |
| GFP-humanCentrin1 | Azimzadeh et al (2009) (Journal of Cell Biology) | Dr A. Delouvée (Institut Curie, Paris, France) |
| Centrin1-VCA-GFP and Centrin1-GFP | Obino et al (2016) (Nature Communications) | Dr P. Pierobon (Institut Cochin, Paris, France) |
| TEpacVV (H187) | Klarenbeek J et al (2015) (PLoS ONE) | K. Jalink (Netherlands Cancer Institute) |
| ExRaiAKAR2 | Addgene | Cat #161753 |

| Reagent/resource | Reference or source | Identifier or catalog number |
|---|---|---|
| **Antibodies** | | |
| Mouse anti-pericentrin | Abcam | Cat #28144 |
| Rabbit anti-pericentrin | Abcam | Cat #4448 |
| Rabbit anti-Arp2 | Abcam | Cat #47654 |
| Mouse anti-AKAP450 | Becton Dickinson | Cat #611518 |
| Mouse anti-CXCR4 human | R&D Systems | Cat #MAB172 |
| Mouse anti-α tubulin | Sigma | Cat #T9026 |
| Mouse anti-PKA Cα | Becton Dickinson | Cat #610981 |
| AlexaFluor568 donkey anti-mouse | Life Technologies | Cat #A10037 |
| AlexaFluor647 donkey anti-rabbit | Life Technologies | Cat #A32787 |
| AlexaFluor plus 647 donkey anti-mouse | Life Technologies | Cat #A31573 |
| **Chemicals, enzymes, and other reagents** | | |
| RPMI 1640 GlutaMax | Gibco | Cat #61870010 |
| Fetal Calf Serum (FCS) | GIBCO | Cat #A5256801 |
| Penicillin and streptomycin | Eurobio | Cat #CABPES01-OU |
| Amaxa Nucleofactor | Lonza | Cat #VCA-1003 |
| Human AB serum | Biowest | Cat #S4190–100 |
| Pan T cell Isolation Kit | Miltenyi | Cat #130-096-535 |
| recombinant human SDF1-α | Peprotech | Cat #300-28 A |
| VCAM-1 (CD106 Fc chimera protein) | R&D Systems | Cat #862-VC-100 |
| Fura-2/AM | Molecular Probes | Cat #F1225 |
| BAPTA-AM | Thermo Fisher | Cat #B6769 |
| Hoechst 33342 | Molecular Probes | Cat #H1399 |
| AlexaFluor488 conjugated phalloidin | Life Technologies | Cat #A12379 |
| AlexaFluor568 conjugated phalloidin | Life Technologies | Cat #A12380 |
| SiRActin | TebuBio | Cat #SC001 |
| H89 | Calbiochem | Cat #371963 |
| Rp-8-CPT-cAMPS | Biolog | Cat #C011 |
| Ht31 (RRRRRRRRRLIEEAASRIVDAVIEQV) | ProteoGenix | |
| Ht31-P (RRRRRRRRRLIPEAASRPVDAVIEQV) | ProteoGenix | |
| Fluorsave | Millipore | Cat #345789 |
| Transwell | Corning | Cat #3421 |
| **Software** | | |
| Metafluor | Meta Imaging (Roper Scientific) | |
| Metamorph | Meta Imaging (Roper Scientific) | |
| ImageJ / Fiji | https://imagej.net/ | |
| Jamovi software | https://www.jamovi.org/ | |
| Prism | GraphPad | |

## Cell culture and PBT purification

CCRF-CEM, a human T lymphoblast cell line isolated from a patient with lymphoblastic leukemia, were cultured in RPMI 1640 GlutaMax (Gibco, #61870010), supplemented with 10% FCS, 5 U/ml penicillin, and 50 µg/ml streptomycin (Eurobio CABPES01-OU). When specified, cells were transfected by nucleofection (Amaxa Nucleofactor, Lonza VCA-1003) with 5 µg DNA for 5 million cells using the C-016 program. The cells were used the day after nucleofection.

Peripheral blood samples from healthy adult individuals were obtained from EFS (Etablissement Français du Sang) under agreement 18/EFS/030, including informed consent for research purposes. Blood samples were anonymous and not associated with any personal data (including age, gender, or sex). Human peripheral blood mononuclear cells (PBMCs) were isolated using Ficoll gradient (800 × g for 20 min, RT), and T cells were purified using Pan T cell Isolation Kit (Miltenyi, 130-096-535). Human T cells cultured in RPMI 1640 GlutaMax supplemented with 10% Human AB serum (Biowest, #S4190–100), 50 U/ml penicillin, and 50 µg/ml streptomycin (Eurobio CABPES01-OU) were used for experiments the day after purification.

## Reagents

CXCL12 (recombinant human SDF1-α) was purchased from Peprotech (300-28 A) and VCAM-1 (CD106 Fc chimera protein) from R&D Systems (862-VC-100). Calcium measurements were performed with Fura-2/AM (Molecular Probes, F1225). For calcium chelation, cells were incubated with 20 µM BAPTA-AM (Thermo Fisher B6769) for 30 min. Nucleus labeling was performed for 4 min with 2 µg/ml Hoechst 33342 (Molecular Probes, H1399). F-actin detection was performed by expressing the LifeAct-mCherry construct (gift from Dr. A. Benmerah) or with Alexa-Fluor488 or AlexaFluor568 conjugated phalloidin (Life Technologies, A12379 or A12380, 1/1000). Stable actin detection was performed with SiRActin (TebuBio, SC001) as previously described (Simao et al, 2021). GFP-Centrin1 was a gift from Dr. A. Delouvée (Institut Curie, Paris, France) and GFP-Centrin1-VCA and GFP-Centrin1 were obtained from Dr. Paolo Pierobon (Institut Cochin, Paris). For cAMP measurements, cells were transfected with the most sensitive version of TEpacVV (H187), which was a gift from Dr. K. Jalink (Netherlands Cancer Institute). Monitoring of PKA activity is achieved through the expression of the ExRaiAKAR2 construct (Addgene, #161753). For PKA activity inhibition, cells were pretreated with 10 µM of H89 (Calbiochem, 371963), 200 µM Rp-8-CPT-cAMPS (Biolog, C011) for 30 min. For PKA delocalization from type II AKAPs, cells were pretreated with 20 µM of the inhibitory peptide Ht31 (RRRRRRRRRLIEEAASRIVDAVIEQV) or the control peptide Ht31-P (RRRRRRRRRLIPEAASRPVDA-VIEQV) for 1 h (synthetized by ProteoGenix).

The following primary antibodies were used for immunofluorescence: mouse anti-pericentrin (Abcam, #28144, 1/1000), rabbit anti-pericentrin (Abcam, #4448, 1/2000), rabbit anti-Arp2 (Abcam, #47654, 1/200), anti-AKAP450 (Becton Dickinson 611518, 1/50), Mouse anti-CXCR4 human (R et D Systems, #MAB172, 1/200), Mouse anti-α tubulin (Sigma T9026, 1/1000), Mouse anti-PKA Cα

(Becton Dickinson 610981, 1/50). Secondary antibodies were the following: AlexaFluor568 donkey anti-mouse, AlexaFluor647 donkey anti-rabbit, and AlexaFluor plus 647 donkey anti-mouse (Life Technologies, A10037, A32787, and A31573, 1/1000 or 1/500).

## Cell stimulation and labeling

For cell roundness measurements, cells were deposited on coverslips coated with 1 µg/ml VCAM-1 and, when specified, 1 µg/ml CXCL12 (except for Fig. 2G, 0.3 µg/ml) and kept at 37 °C for the indicated time. This type of stimulation allows the cells to polarize in the plane of the coverslip and thus to quantify the deformation. For immunofluorescence experiments, cells were deposited on coverslips coated with 1 µg/ml VCAM-1 and, after 5 min of adhesion, 100 ng/ml CXCL12 (except for Fig. 3D, 25 ng/ml) was added to the medium, allowing synchronized stimulation and therefore more precise measurements of the kinetics.

Cells were fixed with 4% paraformaldehyde for 20 min at room temperature. Cells were then permeabilized for 20 min with PBS-BSA-Saponin (1x/0.5%/0.1%), incubated for 45 min with the primary antibody, and 45 min more with the secondary antibody and fluorescent phalloidin. For CXCR4, the labeling was performed on unpermeabilized cells.

For Tubulin and PKA staining, cells were fixed with 4% PFA + 0.2% Glutaraldehyde in PHEM solution (60 mM PIPES, 25 mM HEPES, 10 mM EGTA, 2 mM Mg acetate) + 0.5% Triton, for 12 min. Cells were then permeabilized for 20 min with PBS-BSA-Saponin (1x/0.5%/0.1%), incubated for 45 min with the primary antibody and 45 min more with the secondary antibody and fluorescent phalloidin.

Coverslips were mounted with Fluorsave (Millipore, #345789). Microscopy images were acquired with a wide-field microscope, Nikon Eclipse TE2000-U, and a CMOS camera (ORCA-flash4.0 LT, Hamamatsu). Images were acquired with Metamorph software with a 20x/0.75 objective (morphology measurements) or 60x or 100x/0.5–1.3 oil immersion objective (fluorescence quantification).

## Time-lapse imaging and analysis

Live imaging experiments were performed at 37 °C with a wide-field Nikon TE2000, equipped with a CMOS camera (ORCA-flash4.0 LT, Hamamatsu). Images were acquired every 10 s with Metafluor software. For monitoring Ca, cAMP, PKA activity, and centrosomal actin, a 40x/1.30 oil immersion objective was used. The dynamic imaging experiments are all performed in saline medium (140 mM NaCl, 5 mM KCl, 1 mM CaCl$_2$, 1 mM MgCl$_2$, 11 mM glucose, 10 mM Hepes; pH 7.4).

In all figures and for each individual cell, traces have been normalized to the ratio at the time of the stimulation (R/R0) before averaging.

### cAMP

cAMP measurements were performed with the FRET biosensor TEpacVV. Briefly, when cAMP increases, the probe undergoes a conformational change that allows a decrease of energy transfer between the turquoise fluorophore and the two Venus molecules. Three images were acquired every 10 s: visible, Turquoise channel, and FRET channel. The ratio $R = I_{436/470}/I_{436/535}$, which gives an estimate of cAMP concentration, was calculated with MetaFluor (Roper Scientific) after background subtraction. An increase in this ratio corresponds to an increase in cAMP concentration.

### PKA activity

cAMP measurements were performed with PKA biosensor ExRaiA-KAR2 possessing a circularly permuted GFP (cpGFP). Upon PKA phosphorylation of the probe, a conformation change shifts the maximum excitation wavelength from 400 to 480 nm. Three images were acquired every 10 s: visible, Exc 480 nm/Em 515 nm and Exc 405 nm/Em 515 nm. The ratio $R = I_{480/515}/I_{405/515}$ is calculated with MetaFluor after subtraction of the background noise. An increase in this ratio corresponds to an increase in PKA activity.

### Simultaneous Ca and cAMP measurements

Cells transfected with TEpacVV were loaded with Fura-2/AM at a low concentration (50 nM) in order to avoid interferences between the different fluorophores. Five images were acquired every 10 s: visible, Exc 340 nm/Em 510 nm, Exc 380 nm/Em 510 nm, Exc 436 nm/Em 470 nm, Exc 436 nm, Em 535 nm. The ratios $R = I_{340/510}/I_{380/510}$ and $R = I_{436/470}/I_{436/535}$, which give an estimate of Ca and cAMP concentrations respectively, were calculated with MetaFluor (Roper Scientific) after background subtraction.

## Image analysis

### Cell roundness

The cell roundness parameter was quantified with ImageJ software and corresponds to: $4\text{area}/\pi(\text{major axis})^2$. It is equal to 1 for a round cell and <1 for a deformed one.

### Centrosome position quantification

CEM cells transfected with GFP-centrin1 and labeled with SiRActin were deposited on a VCAM-1-coated coverslip for 5 min before adding CXCL12. The images were acquired every 10 s. The image corresponding to the opening of the actin cortex was determined by watching movies and considered only relative to the previous and the next ones. Indeed, we first determined the period when (1) SiRActin labeling stably decreases at one pole of the cell and (2) the cell simultaneously deforms (transmitted light pictures). Playing the sequence backward, allowed us to determine the beginning of the cortex opening. At this time, we collected the coordinates of the centrosome, the cell center, and the actin cortex opening. Based on this information, we calculated the angle between the two lines [Center of the cell - Center of the opening of the cortical actin cage] and [Center of the cell - Centrosome].

### Centrosomal actin and Arp2 quantification

Accumulation of Actin or Arp2 around the centrosome is quantified by measuring integrated intensities of the labeling in a circle of 1.5-µm radius, centered on the centrosome (pericentrin or centrin labeling). A background value is subtracted from this measurement, corresponding to the integrated intensity measured in a similar circle, designed in the cell but outside of the centrosome area.

### Quantification of centrosomal PKA Cα accumulation

Accumulation of PKA Cα at the centrosome is quantified by calculating the ratio between integrated intensities of the labeling in a circle of 1.5-µm radius, centered on the centrosome, with the one

measured in a circle of 1.5-µm radius positioned in the cytoplasm. A ratio superior to 1 means that the protein is enriched around the centrosome.

### Pearson coefficient

Pearson's coefficient was measured on ImageJ by using a macro containing the Coloc2 plugin. This coefficient measures the degree of overlap between two stainings: perincentrin and AKAP450. A Pearson Coefficient value of 0 means that there is no colocalization between the two stainings. Conversely, a Pearson Coefficient value of 1 means that there is a perfect colocalization.

### Migration assay

The migration assay was performed as previously described (Megrelis and Delon, 2014). Briefly, 300 µl of CEM cells (1 M/ml, complete medium) were deposited in the upper compartment of a Transwell insert (pore diameter: 5 µm; Corning #3421). When indicated, the cells were preincubated with 10 µM H89 for 30 min. The lower compartment contained no chemokine or 200 ng/ml CXCL12 in 600 µl of complete medium. In H89 conditions, the drug was also added to the two compartments during the migration assay. After 2 h of incubation, the inserts were removed, an equal amount of 10 µl calibration beads (Flow-Check Fluorospheres, Beckman Coulter) was deposited in the wells, and the number of cells having migrated to the lower compartment relative to the number of beads was quantified by flow cytometry (FACS Calibur, BD). In parallel, 300 µl of CEM cells (1 M/ml, complete medium) were incubated for 2 h in wells without an insert containing 600 µl of complete medium. 10 µM H89 were added when indicated. The total amount of cells in these wells was then measured through the same method. Migration was quantified in each condition as the number of migrating cells/total cell number x 100.

### Statistics

Each experiment was repeated $N$ times, as specified in the figure legends, along with the number of cells analyzed in each experiment for a given condition. Data were formatted with Prism Software. In the paper, mean ± SE are indicated. Statistical comparisons were performed with Jamovi software through $t$-test (Fig. 2C), paired $t$-test (Fig. EV5B), or nonparametric ANOVA test for multiple comparisons (Friedman for paired experiments or Kruskal–Wallis for unpaired ones). Two-way ANOVA was used for curve comparisons. $*p < 0.05$, $**p < 0.01$, $***p < 0.001$, $****p < 0.0001$.

## Data availability

This study includes no data deposited in external repositories. The data that support the findings of this study are available from the corresponding author upon request.

Expanded View for this article is available online.

The source data of this paper are collected in the following database record: biostudies:S-SCDT-10_1038-S44319-025-00533-2.

## Peer review information

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

## Acknowledgements

We thank Luca Simula for human primary T cells purification, G. Pidoux for helpful discussions on AKAP, J. Delon and P. Pierobon for helpful discussions and comments on the manuscript. This work was supported by CNRS, Inserm, and Université Paris Cité. M.S. was supported by the Ministère de l'Enseignement Supérieur et de la Recherche and by the Labex Who Am I?.

## Author contributions

**Morgane Simao**: Conceptualization; Formal analysis; Validation; Methodology; Writing—original draft; Writing—review and editing. **Fabienne Régnier**: Conceptualization; Formal analysis; Validation; Methodology; Writing—original

draft; Writing—review and editing. **Clotilde Randriamampita**: Conceptualization; Formal analysis; Supervision; Funding acquisition; Validation; Methodology; Writing—original draft; Project administration; Writing—review and editing.

In addition to the CRediT author contributions listed above, the contributions in detail are:

MS, FR, and CR designed and performed experiments and analyzed data. MS and CR wrote the manuscript.

Source data underlying the figure panels in this paper may have individual authorship assigned. Where available, figure panel/source data authorship is listed in the following database record: biostudies:S-SCDT-10_1038-S44319-025-00533-2.

## Disclosure and competing interests statement

The authors declare no competing interests.

# Expanded View Figures

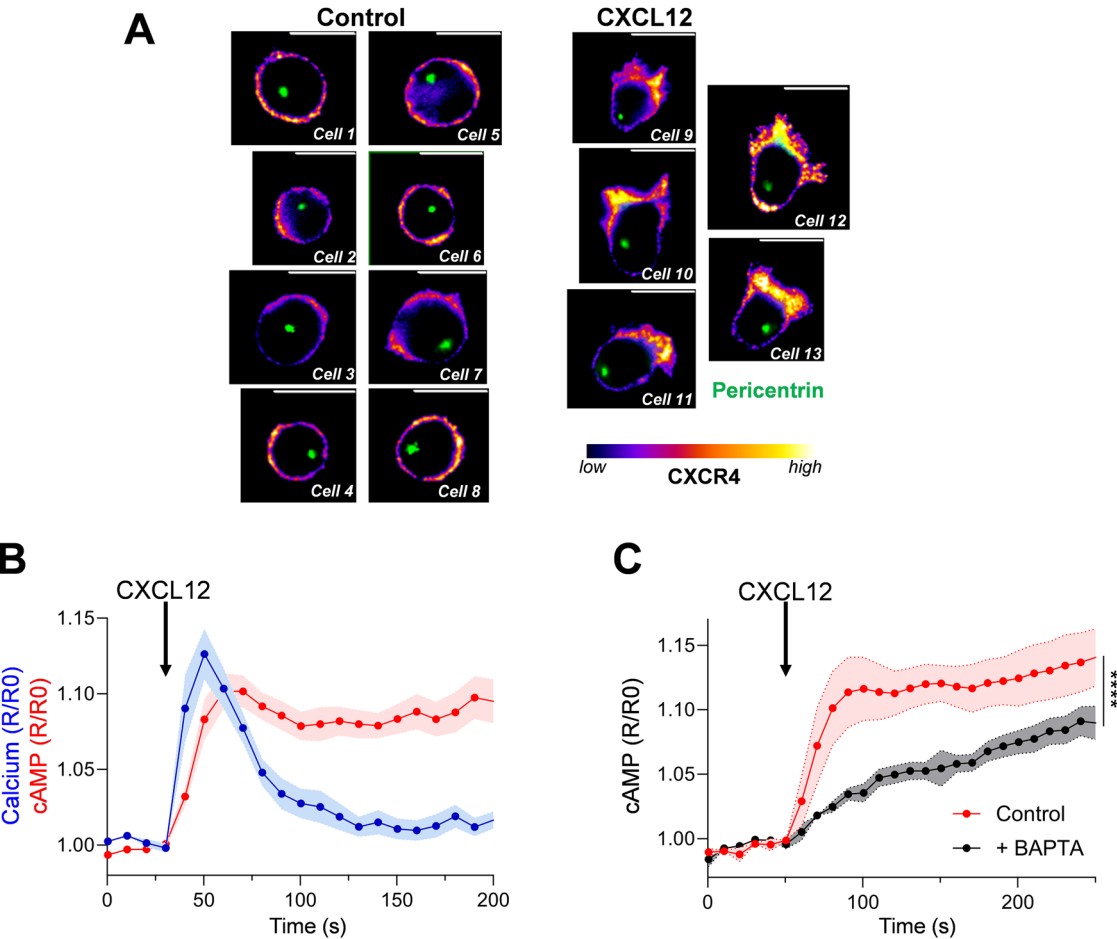

**Figure EV1. Distribution of CXCR4 and variations of Calcium and cAMP upon CXCL12 stimulation.**

(A) Example of CXCR4 distribution revealed by immunofluorescence in eight different unstimulated (left) and five different CXCL12-stimulated (right) CEM T cells. The centrosome is labeled with an antibody against pericentrin (green). Scale bar = 10 μm. (B) Simultaneous measurements of Ca and cAMP variations in CEM T cells transfected with TEpacVV and loaded with Fura-2. CXCL12 stimulation is indicated by the black arrow. For both parameters, values have been normalized relative to the value at the time of stimulation. Mean ± SE of 13 cells. (C) CXCL12-induced cAMP variations in CEM T cells transfected with TEpacVV and preincubated or not with the Ca chelator BAPTA/AM. Values have been normalized relatively to the Ca value at the time of stimulation. Mean ± SE of five independent experiments, 6–19 cells/condition/experiment. Statistics: Two-way Anova. ****$p < 0.0001$. Source data are available online for this figure.

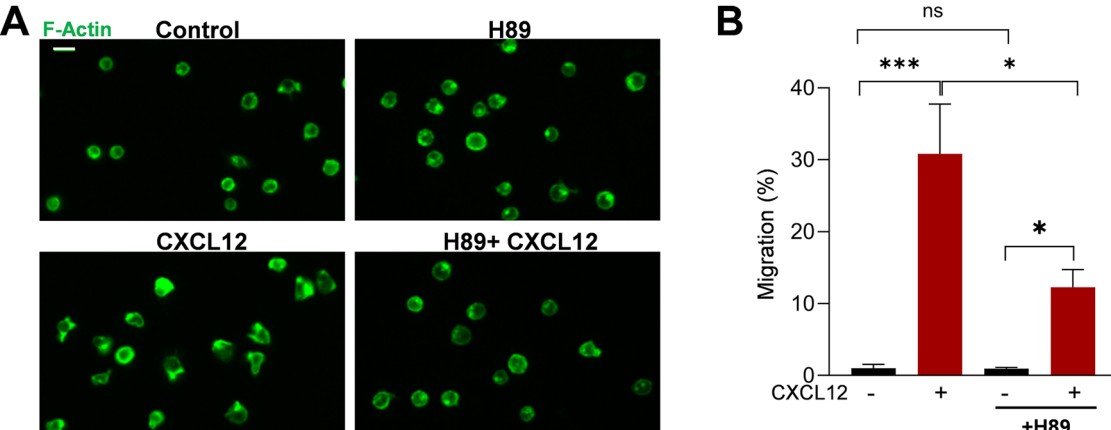

**Figure EV2. Involvement of PKA in CXCL12-induced CEM T cell deformation and migration.**

(A) Example of morphology of CEM T cell stimulated or not with CXCL12 and with or without pretreatment with H89. F-Actin (green) has been labeled with Phalloidin. Scale bar = 20 μm. (B) Transwell migration assay with CEM T cells pretreated or not with H89. When indicated, CXCL12 was added to the lower compartment. Mean ± SE of three independent experiments. Statistics: Nonparametric ANOVA test for multiple comparisons. \*\*\*$p \leq 0.001$, \*$p < 0.041$. Source data are available online for this figure.

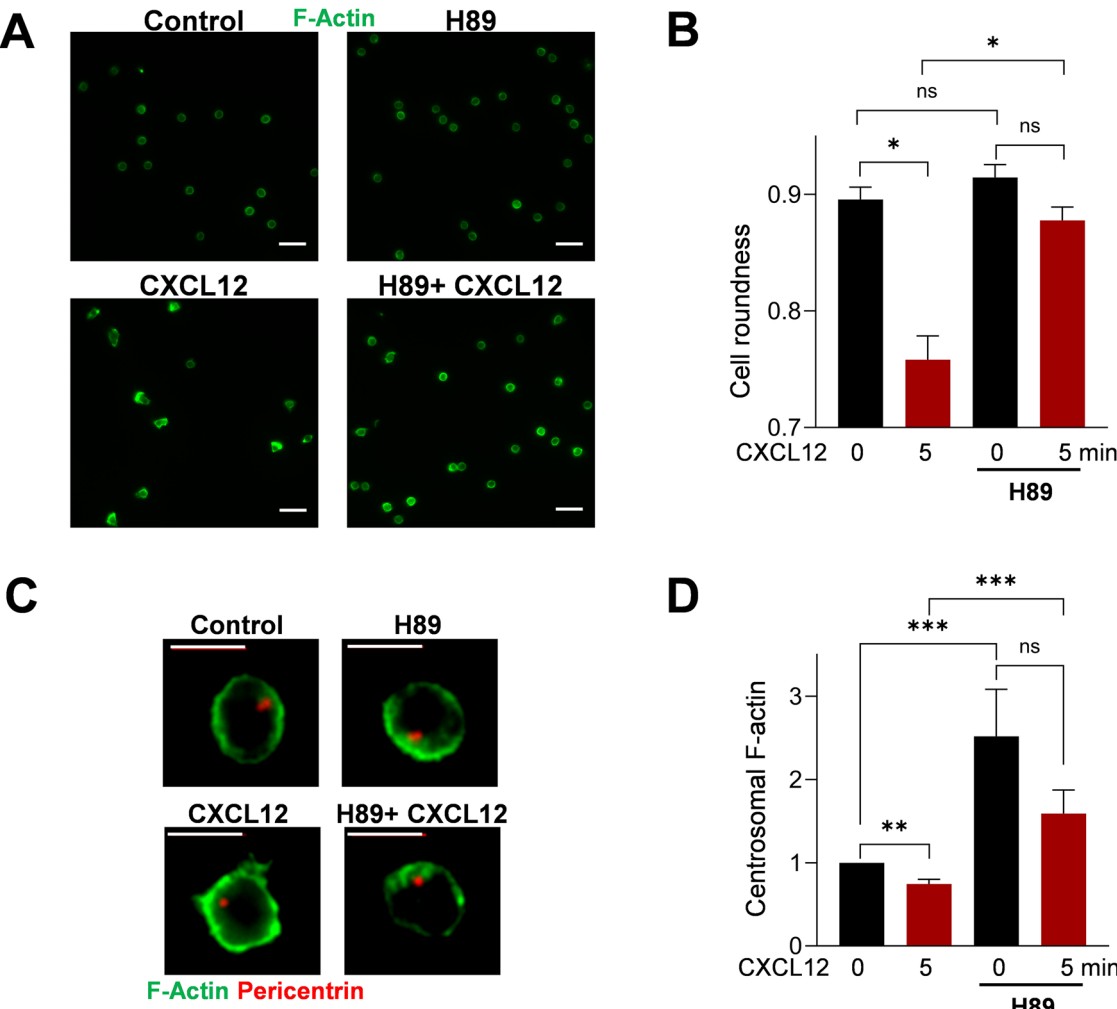

**Figure EV3. Sensitivity of CXCL12-stimulated human primary blood T cells to PKA inhibitor.**

(A) Example of morphology of PBT stimulated or not with CXCL12 and with or without pretreatment with H89. F-Actin (green) has been labeled with Phalloidin. Scale bar = 20 μm. (B) Quantification of cell roundness of PBT stimulated or not with CXCL12 and with or without pretreatment with H89. Mean ± SE of four independent experiments (four different donors), 64–871 cells/condition/experiment. Statistics: Nonparametric ANOVA test for multiple comparisons. *$p = 0.017$. (C) Example of F-actin distribution in PBT stimulated or not with CXCL12 and with or without pretreatment with H89. F-actin is labeled with Phalloidin (green) and centrosome with an antibody against pericentrin (red). Scale bar = 20 μm. (D) Quantification of centrosomal actin in PBT stimulated or not with CXCL12 and with or without pretreatment with H89. For each experiment, values have been normalized relative to the mean intensity of centrosomal F-actin measured in unstimulated and untreated PBT. Mean ± SE of four independent experiments (four different donors), 11–57 cells/condition/experiment. Statistics: Nonparametric ANOVA test for multiple comparisons. ***$p < 0.001$, **$p = 0.007$. Source data are available online for this figure.

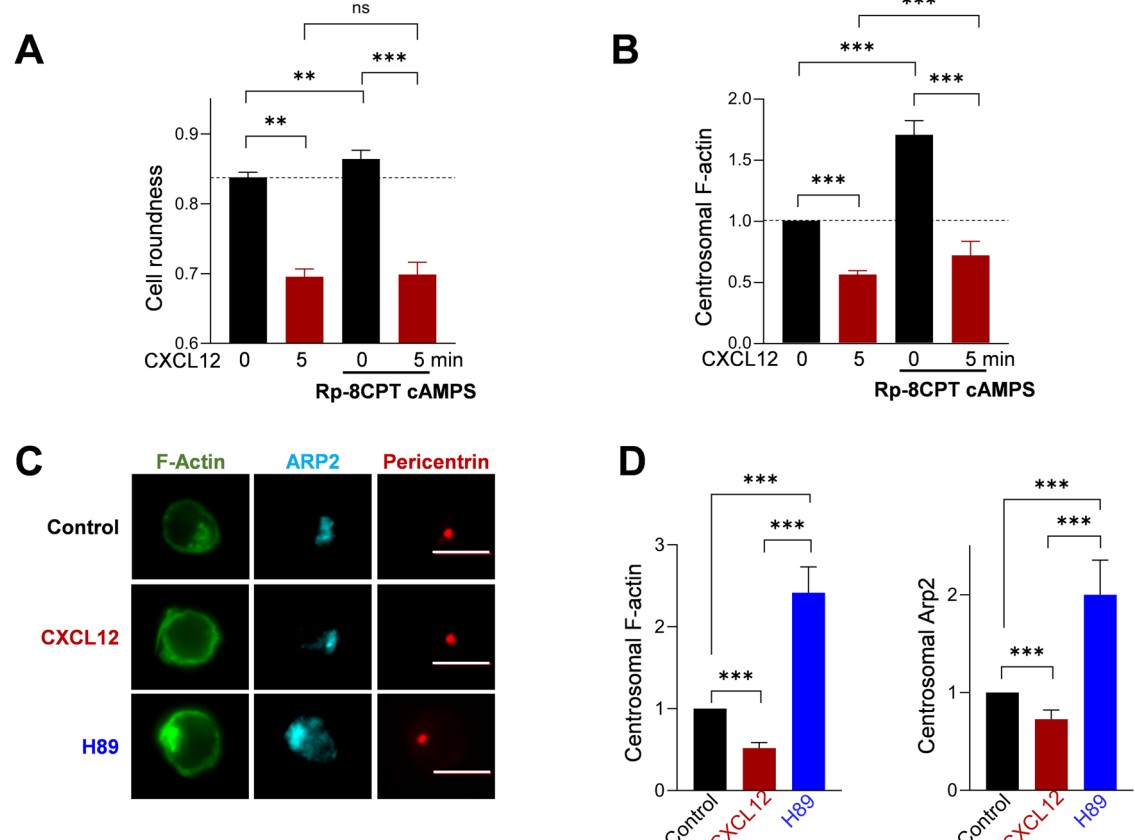

**Figure EV4.** Effect of Rp-8-CPT-cAMPS on cell roundness and centrosomal actin level and centrosomal Arp2 and F-Actin distribution upon CXCL12 and PKA inhibition.

(A) Cell roundness of CEM T cells pretreated or not with Rp-8-CPT-cAMPS and deposited on VCAM-1 or VCAM-1 + CXCL12 (1 µg/ml) for 5 min. Mean ± SE of four independent experiments, 500–1226 cells/condition/experiment. Statistics: Nonparametric ANOVA test for multiple comparisons. ***$p < 0.001$, **$p \leq 0.007$. (B) Centrosomal actin was measured in CEM T cells pretreated or not with Rp-8-CPT-cAMPS and stimulated or not with CXCL12 (100 ng/ml) for 5 min. For each experiment, values have been normalized relative to the mean intensity of centrosomal F-actin measured in unstimulated and untreated CEM T cells. Mean ± SE of three independent experiments, 26–61 cells/condition/experiment. Statistics: Nonparametric ANOVA test for multiple comparisons. ***$p < 0.001$ (C) Example of F-Actin and Arp2 distributions in representative CEM T cells in control conditions (upper line), after CXCL12 stimulation (middle line) or after H89 pretreatment (lower line). Scale bar = 10 µm. (D) Effect of CXCL12 stimulation and PKA inhibition (H89) on F-Actin and Arp2/3 intensities at the centrosome. The quantification of the two proteins have been performed on the same cells. For each experiment and for both proteins, values have been normalized relative to the mean intensity measured in unstimulated and untreated CEM T cells. Mean ± SE of four independent experiments, 38–165 cells/condition/experiment. Statistics: Nonparametric ANOVA test for multiple comparisons. ***$p < 0.001$. Source data are available online for this figure.

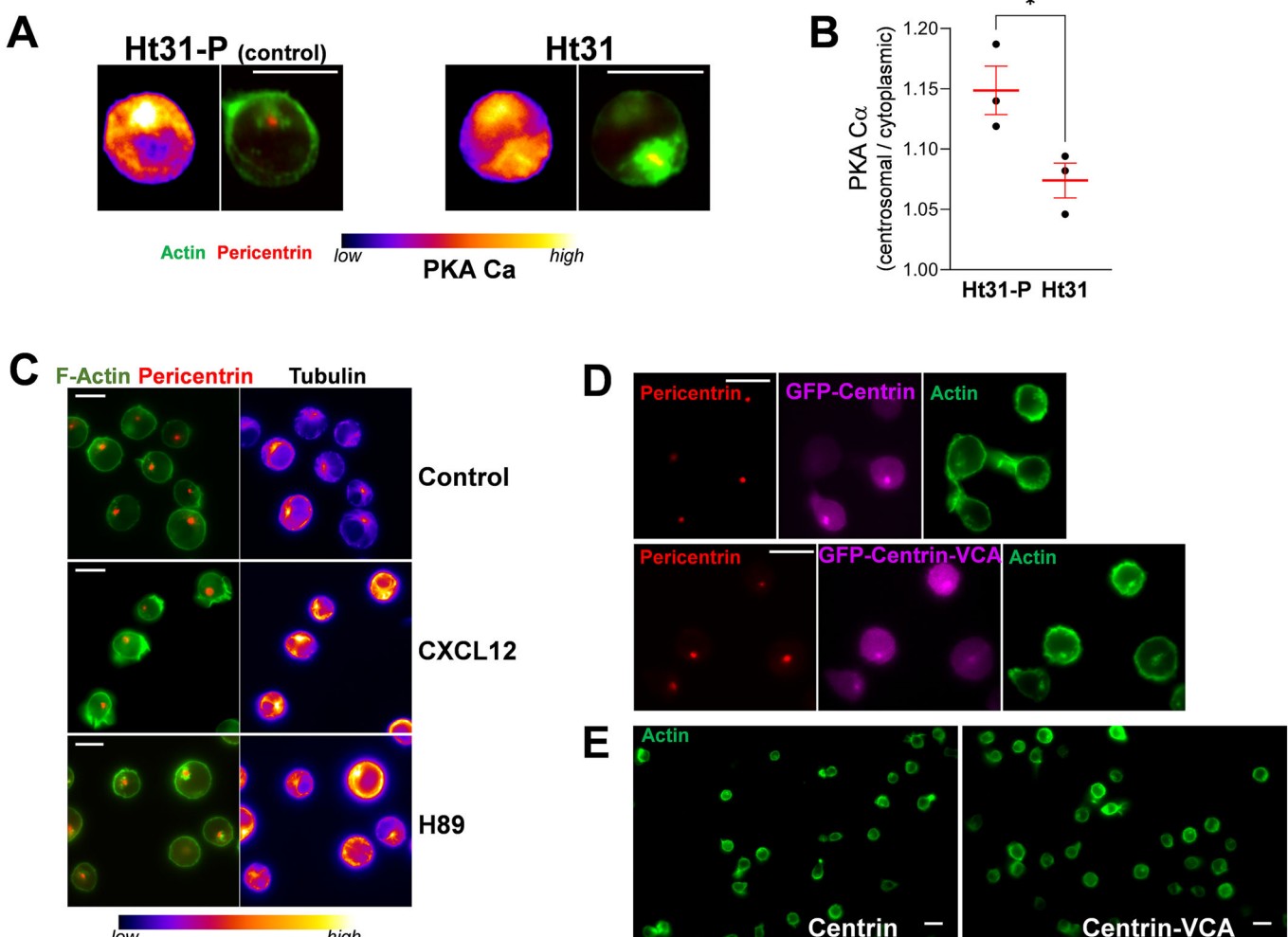

**Figure EV5.  PKA catalytic subunit, microtubules, and actin distributions.**

(A) Example of PKA Cα distribution in CEM T cells treated with control peptide (Ht31-P) or with AKAP II inhibitor, Ht31. F-Actin labeled with phalloidin (green) and centrosome revealed with an antibody against pericentrin (red). Scale bar: 10 μm. (B) The ratio between PKA Cα intensities measured in a 3 μm diameter area centered at the centrosome with the ones measured in a similar area in the cytoplasm in cells either treated with the control peptide (Ht31-P) or with the AKAP II inhibitor (Ht31). A ratio >1 reflects an accumulation of the protein around the centrosome. Mean ± SE of three independent experiments, 12–95 cells/condition/experiment. Statistics: Paired $t$-test *$p = 0.0484$. (C) Examples of cytoskeleton distribution in CEM T cells in control conditions, stimulated with CXCL12, or pretreated with H89. F-Actin (green) is labeled with phalloidin, the centrosome with anti-pericentrin antibody, and microtubules with anti-α tubulin antibody. Scale bar = 10 μm. (D) Example of F-actin distribution (green) in CEM T cells transfected with either GFP-centrin1 or with GFP-centrin1-VCA (magenta). The centrosome is revealed with an antibody against pericentrin (red). Scale bar = 10 μm. (E) Example of morphology of CEM T cell transfected with either GFP-centrin1 or with GFP-centrin1-VCA. F-actin is labeled with phalloidin (green). Scale bar: 10 μm. Source data are available online for this figure.

