## [Peer Review File · EMBO Reports]

Centrosomal actin pool levels regulated by localized PKA set the threshold for T cell polarization

Morgane Simao, Fabienne Régnier, and clotilde randriamampita

Corresponding author(s): clotilde randriamampita (clotilde.randriamampita@inserm.fr)

Review Timeline:

Transfer Date:	24th Apr 25
Editorial Decision:	22nd May 25
Revision Received:	4th Jun 25
Accepted:	7th Jul 25

Editor: Deniz Senyilmaz Tiebe

Transaction Report: A revised version of this manuscript was transferred to EMBO reports following peer review at

**Review
COMMONS**

Review Commons and The EMBO Journal.

Review #1

1. Evidence, reproducibility and clarity:

Evidence, reproducibility and clarity (Required)

The authors study early events initiated in T cells upon chemokine activation leading to cell polarization preceding cell migration. They show that actin dynamics at the centrosome are regulated in a cAMP/PKA dependent manner using different reporter systems and imaging approaches

2. Significance:

Significance (Required)

This work will be of interest to immunologists with a strong cell biology background. The strength of the study is the detailed cell biological / near biophysical analysis of early changes in actin dynamics in relation to centrosome positioning. The data is well controlled and convincing. Conclusions adequate based on available data. The limitation I see is the use of a single cell line system of cancerous origin and the fact that only changes in cellular morphology are quantified, but not cellular behavior itself - e.g. migration, T cell intrinsic signalling. If some key observations can be validated in primary T cells this would be perfect, or at least in a second model system. If signalling related to changed morphology is affected by regional inhibition of PKA/actin remodelling, remains uncertain, too - maybe there is a way to monitor additional parameters, other than roundness/PKA activity

3. How much time do you estimate the authors will need to complete the suggested revisions:

Estimated time to Complete Revisions (Required)

(Decision Recommendation)

Between 1 and 3 months

4. Review Commons values the work of reviewers and encourages them to get credit for their work. Select 'Yes' below to register your reviewing activity at Web of Science Reviewer Recognition Service (formerly Publons); note that the content of your review will not be visible on Web of Science.

Yes

Review #2

1. Evidence, reproducibility and clarity:

Evidence, reproducibility and clarity (Required)

The ability of T cells to migrate along chemotactic cues is critical for the initiation and regulation of adaptive immune responses. To migrate directionally, cells require a polarised shape and intracellular organisation, allowing intracellular force generation towards the intended direction of migration.

The manuscript by Simao et al investigates the role of the centrosome and its co-localised pool of the actin cytoskeleton in defining the direction of an initial polarisation while being surrounded by homogenous chemokine concentration. The authors (i) describe a correlation of the intracellular position of the centrosome with the site of polarisation, (ii) identify that a reduced amount of actin at the centrosome is beneficial for cell polarisation and that (iii) the protein kinase PKA regulates this actin pool at the centrosome.

Overall, the data presented appear convincing but would benefit from a more detailed presentation of representative image examples and additional experimental data.

****Major points:****

(i) Many experimental microscopy datasets are quantified but lack representative images. These representative images are important to be able to judge the underlying data and should be included for the datasets shown in Figure 2, Fig. 3c, Fig. 4d (Lifeact examples), and Fig. 6 (Centrin VCA). In addition, the differences in the signal intensity of the actin cytoskeleton are sometimes hardly visible in the provided representative images (e.g., Fig. 4b, control vs CXCL12). Could the authors come up with solutions to show this in a better way (e.g., zooms and/or fire-colour coding)?

(ii) The authors use an experimental setup in which the cells 'see' a homogenous chemokine concentration around them. Further, they discuss different models in the introduction of how the local cellular polarisation in such a uniform chemokine sounding is defined, including a model of polarised localisation of chemokine receptors on the plasma membrane. However, in a tissue, chemokines are typically not homogeneously distributed

but are present in the form of a gradient, e.g. due to local chemokine sources or the self-generation of chemotactic gradients by neighbouring migrating cells. Therefore, it would be interesting to know whether the described repositioning of the centrosome and the changes in the actin pool are also important if cells are in a chemokine gradient.

Experimentally, this could be addressed by providing a local chemokine source (e.g. from a micropipette). If this goes beyond the scope of the manuscript, then at least this aspect of chemokine gradients should be clearly mentioned and discussed in the introduction and the discussion sections.

(iii) Given that the centrosome acts as a microtubule-organising center (MTOC), it would be interesting to see the microtubules during polarisation. Did the author try the visualisation of microtubules in live or by immunofluorescence stainings? Would it be a plausible model that reduced actin polarisation allows microtubule polarisation towards the cell periphery and thereby induces a protrusion by delivering signalling and cytoskeleton components towards the newly forming protrusion? May this be only targeted towards one side of the centrosome, as the nucleus may sterically hinder the efficient growth of microtubules to the other cellular side? What would happen in cells without a nucleus? And what would happen in cells without a centrosome (e.g., by PLK4 inhibition via centrinone) - are cells still able to form protrusions efficiently? These experimental suggestions are optional but could significantly improve the study.

(iv) Bringing actin artificially to the centrosome via the centrin-VCA construct is a very nice approach. However, the dataset would strongly benefit from samples, in which the chemokine CXCL12 is included.

****Minor points:****

(i) The study is based on a lymphoblastic cell line called CEM T cells. This should be clearly stated at the beginning of the results section and in all figure legends, as it remains unknown whether primary T lymphocytes would show the same behaviour. Additionally, the methods section should contain more details about this cell line, e.g. whether it is from mouse or human origin.

(ii) The authors mention 'suboptimal' conditions of stimulation. However, it remains unclear in the results sections what this means. Some of the experimental modulations (e.g. the cAMP analog Rp-8-CT-cAMPS) seem to only show an effect in these suboptimal conditions but not in the optimal conditions. This should be clearly stated and discussed.

(iii) 23 out of the 38 references are older than 5 years, and most of them are older than 10 years. While it is surely very important to refer to these older findings, the authors may include more knowledge from recent years. This may include references about centrosome positioning in immune cells and motile cells (e.g., PMID32379884, PMID29934494, PMID30944468, PMID37987147, PMID36398880, PMID38627564, PMID33634136) and the actin cytoskeleton at the centrosome (PMID33609453, PMID33184056, PMID36111670).

(iv) The first paragraph of the discussion (lines 253 - 262) needs references.

(v) Some Figures maybe combined as the findings are closely related (e.g., Figures 2 and 3; and Figures 5 and 6).

(vi) Line 94, Supply. Fig. 1: the authors that the chemokine receptor CXCR4 has a uniform distribution in non-stimulated cells. This is not directly evident in the images as there are areas of more and less signal. It would be important to clearly describe this in the text. Further, the labelling of the figure would benefit from labelings such as 'cell 1', 'cell 2', etc to directly make clear that these are images from 3 representative cells.

(vii) Different centrin isoforms exist (centrin 1, 2, 3). It should be mentioned in the results and methods section, which isoform was used for their genetic constructs (e.g., centrin-GFP, centrin-VCA).

2. Significance:

Significance (Required)

This manuscript employs a lymphoblastic cell line called CEM T cells as a model for T lymphocytes. Using imaging of these cells on 2D substrates with and without chemokine, the authors identify a PKA-controlled actin pool at the centrosome that appears to regulate the local site of protrusion formation during cell polarisation. This is an interesting finding that adds to the knowledge of (i) the functions of centrosome positioning and (ii) the functions of the actin cytoskeleton at the centrosome. Thus, the study will be interesting to readers in the centrosome and migration fields. To broaden the scope of the manuscript, the findings could be tested in primary T lymphocytes and mechanistically address the role of microtubules within the described process.

3. How much time do you estimate the authors will need to complete the suggested revisions:

Estimated time to Complete Revisions (Required)

(Decision Recommendation)

Between 3 and 6 months

4. Review Commons values the work of reviewers and encourages them to get credit for their work. Select 'Yes' below to register your reviewing activity at Web of Science Reviewer Recognition Service (formerly Publons); note that the content of your review will not be visible on Web of Science.

Yes

Review #3

1. Evidence, reproducibility and clarity:

Evidence, reproducibility and clarity (Required)

****Summary****

Using a variety of live- and fixed-cell imaging techniques, the authors make correlative and causative connections between chemokine-stimulated increases in cAMP and localized PKA activity, positioning and F-actin content of the centrosome, and cell polarity in T-cells.

****Major Comments****

1. The authors state that "uniform CXCR4 labelling is observed in unstimulated cells" (ln94-95). While the panels in Supp Fig1a show a pattern in unstimulated cells that is obviously less dramatically asymmetrical than seen for stimulated (and, importantly, already polarized) cells in Supp Fig1b, the labelling in unstimulated cells is still far from uniform, as there is considerable heterogeneity of signal intensity along the perimeter. This is important, given that they are looking at a membrane receptor, even small fluctuations in which may be greatly amplified and thus have considerable effects on symmetry downstream. The authors should either quantify the intensity (e.g. signal as a function of polar coordinate value) or soften their language to more accurately reflect the data.
2. Regarding the analysis of polarization events in Fig. 1c and 1d, it is not clear how, exactly, the time point for the cortex opening is determined. For example, in the sample images in 1c, would the +110s or the +340s time point be used? The reason this is important is that the angle seems to change with time (at least in the example given) and there are also

heterogeneities (specifically, decreases) in SiR-actin intensity along the cortex that precede cortex opening. Thus, it is not clear whether the cortex begins to open in closer proximity to the centrosome or whether the centrosome is further aligned after the cortex opens.

3. Also regarding Figure 1d, it is not clear how many cells and experimental replicates are represented in the data - the Results text reports 60 cells from 11 experiments (ln106) but the legend reports 58 cells (ln564) without mention of experimental replicate number. Also, while the rose plot is useful, it is important to have statistical analysis on the skewness of the response and/or to report something other than the average angle - for example, the percentage of cells with a cortex opening in the same 90-degree quadrant as the centrosome. Finally, it might be clearer to the reader to have the rose plot and the model cell oriented in the same direction.

4. For their PKA inhibition experiments, the authors introduce H89 as "a competitive inhibitor of ATP on the PKA catalytic subunit" (ln138-139). H89 is a very non-specific inhibitor, as demonstrated by Davies et al (PMID 10998351) and reviewed by Lochner & Moolman (PMID 17214602) and should be introduced more accurately. To their credit, the authors use an orthogonal approach of PKA inhibition with a cAMP analog and see comparable effects. Those data, currently in Supp Fig. 3, should be included as primary data, given that the H89 data can only be correctly interpreted in the context of the Rp-cAMPS data (this applies to both Fig3 and Fig4).

5. As part of Fig.5, the authors state "AKAP450 is a type II AKAP i.e. it is able to bind RII subunits of PKA. In order to determine whether this AKAP allows a [sic] compartmentalization of the PKA activity responsible for centrosomal F-actin regulation, we used the specific peptide (Ht31)". Ht31 broadly inhibits PKA anchoring; its effect is not specific for any individual AKAP, including AKAP9/AKAP450. Moreover, the authors neither show/confirm PKA localization to the centrosome nor its displacement with the indicated concentration of Ht31, and they do not include data that PKA is displaced from the centrosome with any greater specificity or sensitivity than its displacement from any other subcellular location. Therefore, this statement (as well as the claim in the Abstract that "a specific pool of protein kinase A) (ln15) by the authors is not accurate. The authors should, at the very least, re-word the statement and, for the sake of rigor and support of their hypotheses, confirm that PKA (subunits and/or activity) is displaced from the centrosome.

6. The experiments & results using VCA-centrin-GFP are very intriguing. However, it is crucial that primary data (i.e. photomicrographs/panels of fluorescent images of centrin/VCA-centrin localization, centrosomal F-actin, and roundness) be included for the readers' inspection. Also, it is not clear whether the graphically summarized data on centrosomal F-actin and roundness (Fig. 6b) represent analysis of cells before and after CXCL12 stimulation, or only before or only after. If either of the latter, analysis of these

parameters both before and after stimulation should be included.

****Minor Comments****

1. The writing is generally clear and accurate, but often somewhat 'choppy'. As one of many examples: "As depicted in Figure 1b, in this configuration, we observed that cell deformation starts rapidly after stimulation. The mean delay between cell stimulation and the time when cells start to deform is 112 {plus minus} 10 s (n=48 cells)" could be re-written as "As depicted in Figure 1b, in this configuration, we observed that cell deformation starts rapidly after stimulation, with a mean delay between cell stimulation and initiation of cell deformation of 112 {plus minus} 10 s (n=48 cells)." This is completely stylistic, of course, and would simply (albeit slightly) improve the readability of the work.
2. The authors comment that the contribution of calcium, PI3K, and cAMP signaling "in the early processes allowing the establishment of the asymmetric distribution of cellular components and of the polarity regulators is still elusive" (ln33-35) seems a bit overstated, as there have been numerous, impactful contributions investigating each of those pathways.
3. The work seems to start off as being focused on symmetry breaking rather than polarization, but this can be mitigated through rewriting the Introduction.
4. The phrase "the major one" (ln47), presumably referring to one of "several local signaling poles" (earlier in ln47) is ambiguous and should be reworded (e.g. "the pole with the highest density of receptors").
5. The phrase "variations of cAMP after CXCL12 addition upon dynamic cell imaging" (ln112) is not clear.
6. The authors may want to reconsider the use of an ellipsis (ln25), which stands out as somewhat informal for a scientific manuscript.
7. There are several typographical errors throughout that should be addressed (e.g. "AMPC" rather than "cAMP" in the header of Fig7b.; "we were able establish" (ln 130); "while PKA are rapidly activated" (ln133)).
8. The figure legends most often read more like miniature, repeated results sections than detailed descriptions of experimental details and data processing, analysis and depiction.

2. Significance:

Significance (Required)

Directional cell migration is a fundamentally important aspect of cell biology. Understanding the molecular mechanisms that govern cellular symmetry breaking, polarization, and migration are - in turn - important for a fuller understanding of how cells

efficiently move from location to location. T cells, which are highly dependent on efficient and dynamically, cytokine-directed migration for their physiologic function, are an excellent model system in which to unravel such molecular mechanisms. The authors efforts to connect localized cytokine-initiated signaling events with changes in centrosomal actin decoration and thence into cell polarity are, therefore, of considerable potential significance.

3. How much time do you estimate the authors will need to complete the suggested revisions:

Estimated time to Complete Revisions (Required)

(Decision Recommendation)

Between 3 and 6 months

No

Full Revision

Manuscript number: RC-2024-02542

Corresponding author(s): Clotilde, Randriamampita

1. General Statements [optional]
2. Point-by-point description of the revisions

Reviewer #1

Evidence, reproducibility and clarity

The authors study early events initiated in T cells upon chemokine activation leading to cell polarization preceding cell migration. They show that actin dynamics at the centrosome are regulated in a cAMP/PKA dependent manner using different reporter systems and imaging approaches.

Significance

This work will be of interest to immunologists with a strong cell biology background. The strength of the study is the detailed cell biological / near biophysical analysis of early changes in actin dynamics in relation to centrosome positioning. The data is well controlled and convincing. Conclusions adequate based on available data.

The limitation I see is the use of a single cell line system of cancerous origin and the fact that only changes in cellular morphology are quantified, but not cellular behavior itself - e.g migration, T cell intrinsic signalling. If some key observations can be validated in primary T cells this would be perfect, or at least in a second model system. If signalling related to changed morphology is affected by regional inhibition of PKA/actin remodelling, remains uncertain, too - maybe there is a way to monitor additional parameters, other than roundness/PKA activity.

- In order to complete our results, key experiments have been performed in primary human blood T lymphocytes (PBT). 4 experiments have been realized with 4 different donors. In this model, as in CEM T cell line, we have observed that:

- chemokine-induced PBT deformation is inhibited by the PKA inhibitor H89. Unstimulated PBT are poorly deformed so that H89 by itself only slightly (but not significantly) increases resting roundness.

- chemokine stimulation induces a decrease of centrosomal actin which is partly prevented by H89 treatment. In these conditions, centrosomal actin remains at a level superior to the resting one. H89 by itself is sufficient to promote an increase in centrosomal actin.

Altogether, these results demonstrate that in PBT as in CEM, in resting conditions as well as after chemokine stimulation, the level of centrosomal actin is controlled by PKA activity and is associated with cell deformation.

These new results are presented in Fig Supp 4 and described in the manuscript (lines 158-164 and 199-202).

- In order to investigate the consequences of the PKA inhibition on chemokine-induced cell behavior, Transwell migratory assays have been performed. As presented in Fig Supp 3b, the chemokine-triggered T cell migration through 5µm pores is reduced by H89 pretreatment. This suggests that the reduction of T cell deformation after PKA inhibition has a direct consequence on the physiological behavior of T lymphocyte. This result is described in the manuscript (lines 165-169).

Reviewer #2

Evidence, reproducibility and clarity

The ability of T cells to migrate along chemotactic cues is critical for the initiation and regulation of adaptive immune responses. To migrate directionally, cells require a polarised shape and intracellular organisation, allowing intracellular force generation towards the intended direction of migration.

The manuscript by Simao et al investigates the role of the centrosome and its co-localised pool of the actin cytoskeleton in defining the direction of an initial polarisation while being surrounded by homogenous chemokine concentration. The authors (i) describe a correlation of the intracellular position of the centrosome with the site of polarisation, (ii) identify that a reduced amount of actin at the centrosome is beneficial for cell polarisation and that (iii) the protein kinase PKA regulates this actin pool at the centrosome.

Overall, the data presented appear convincing but would benefit from a more detailed presentation of representative image examples and additional experimental data.

Major points:

(i) Many experimental microscopy datasets are quantified but lack representative images. These representative images are important to be able to judge the underlying data and should be included for the datasets shown in Figure 2, Fig. 3c, Fig. 4d (Lifeact examples), and Fig. 6 (Centrin VCA). In addition, the differences in the signal intensity of the actin cytoskeleton are sometimes hardly visible in the provided representative images (e.g., Fig. 4b, control vs CXCL12). Could the authors come up with solutions to show this in a better way (e.g., zooms and/or fire-colour coding)?

As suggested by the reviewer, some representative images have been added for all experiments (Figures 3, Supp 3a, Supp 4 a & c, Supp 7a, Supp 8). Furthermore, a zoom of actin network around the centrosome in different conditions is now presented in Fig 3b (previous Fig 4b)).

(ii) The authors use an experimental setup in which the cells 'see' a homogenous chemokine concentration around them. Further, they discuss different models in the introduction of how the local cellular polarisation in such a uniform chemokine sounding is defined, including a model of polarised localisation of chemokine receptors on the plasma membrane. However, in a tissue,

chemokines are typically not homogeneously distributed but are present in the form of a gradient, e.g. due to local chemokine sources or the self-generation of chemotactic gradients by neighbouring migrating cells. Therefore, it would be interesting to know whether the described repositioning of the centrosome and the changes in the actin pool are also important if cells are in a chemokine gradient. Experimentally, this could be addressed by providing a local chemokine source (e.g. from a micropipette). If this goes beyond the scope of the manuscript, then at least this aspect of chemokine gradients should be clearly mentioned and discussed in the introduction and the discussion sections.

We thank the reviewer for this remark. We had not actually addressed this point.

- We have tried to create chemokine gradients. But, we have never been able to observe all cells polarizing in the same direction (along this gradient). This negative result might however be due to our experimental setup so that this result is not mentioned in the manuscript. However, to our knowledge, no real chemotaxis has been described in T lymphocytes.
- We have performed some Transwell assays that mimic trans-endothelial migration and have shown the involvement of PKA (Figure Supp 3b, lines 165-169). But the involvement of centrosomal actin could not be investigated in this experiment.
- This question is now addressed in introduction (lines 44-47) and in discussion (lines 400-405).

(iii) Given that the centrosome acts as a microtubule-organising center (MTOC), it would be interesting to see the microtubules during polarisation. Did the author try the visualisation of microtubules in live or by immunofluorescence stainings? Would it be a plausible model that reduced actin polarisation allows microtubule polarisation towards the cell periphery and thereby induces a protrusion by delivering signalling and cytoskeleton components towards the newly forming protrusion? May this be only targeted towards one side of the centrosome, as the nucleus may sterically hinder the efficient growth of microtubules to the other cellular side? What would happen in cells without a nucleus? And what would happen in cells without a centrosome (e.g., by PLK4 inhibition via centrinone) - are cells still able to form protrusions efficiently? These experimental suggestions are optional but could significantly improve the study.

- As suggested by the reviewer, we performed new experiments to quantify microtubules together with actin in the centrosomal area. As previously mentioned in discussion, we were expecting that microtubule evolution mirrors that of actin. This is indeed the case after chemokine stimulation, where the quantity of microtubules present in the centrosomal area increases as actin decreases. However, after H89 treatment, an enrichment of both cytoskeleton networks is observed. It suggests that PKA could affect directly microtubule polymerization/depolymerization equilibrium independently of actin level possibly through Microtubules Associated proteins phosphorylation. Nevertheless, the effect of centrosomal actin level on cell polarization we emphasize, might partly be due to its consequence on microtubule growth.

These results are now presented in Figure 4c and representative images shown in Figure Supp 9. Furthermore, this observation is mentioned in the results part (line 256-263) and in discussion (line 428-442).

- The sterical role of the nucleus on the polarization is an interesting hypothesis that we have been however unable to test it. T lymphocytes are small cells (around 15µm diameter for CEM cell line) that seems to be difficult to enucleate.

- We have tried to obtain cells without centrosomes in order to test their ability to polarize. To this purpose, we have treated cells with the PLK4 inhibitor, centrinone (500nM) for 24 to 48h hours. Although we could observe after 33 hours, that the large majority of cells present only one centriole, we never got cells without centrioles. After this one-centriole stage, the cells started dying.

(iv) Bringing actin artificially to the centrosome via the centrin-VCA construct is a very nice approach. However, the dataset would strongly benefit from samples, in which the chemokine CXCL12 is included.

The effect of chemokine on centrin-VCA expressing cells has been investigated. This result is displayed in Fig 4e. In cells expressing the centrin-VCA construct, the level of actin at the centrosome is higher than in control cells. However, upon chemokine stimulation, the signaling pathways leading to actin reduction are still active so that the level of centrosomal actin decreases (insignificantly different to the control) and the cells do not deform (insignificantly different to the control). These results are now discussed in the manuscript (lines 283-287).

Minor points:

(i) The study is based on a lymphoblastic cell line called CEM T cells. This should be clearly stated at the beginning of the results section and in all figure legends, as it remains unknown whether primary T lymphocytes would show the same behavior. Additionally, the methods section should contain more details about this cell line, e.g. whether it is from mouse or human origin.

The cells used for the experiments are now clearly indicated in the figure legends and specified in the manuscript (results and methods).

Furthermore, as mentioned in the response to reviewer #1, new experiments have been performed with primary human blood T cells. Similar effect of PKA on cell deformation and centrosomal actin regulation have been observed. These new experiments are presented in Fig Supp 4 and described in the manuscript (lines 158-164 and 199-202).

(ii) The authors mention 'suboptimal' conditions of stimulation. However, it remains unclear in the results sections what this means. Some of the experimental modulations (e.g. the cAMP analog Rp-8-CT-cAMPS) seem to only show an effect in these suboptimal conditions but not in the optimal conditions. This should be clearly stated and discussed.

In resting cells, Rp-8-CT-cAMPS has a similar effect as H89: it induces an increase in cell roundness as well as in centrosomal actin. However, after chemokine stimulation, the inhibition of cell deformation and of centrosomal actin reduction is only observed with low intensity stimulation (Figures 2g & 3d vs Supp 5a). Our interpretation is that this chemical is not potent enough to counteract strong activation of PKA. Indeed, in resting conditions or after mild PKA activation, it blocks the effect of chemokine. This might be due to the fact that the inhibitor concentration reached within the cells is not enough. The text concerning this part has been modified for more clarity (lines 171-178 and 203-206) and the figure legends mention explicitly the chemokine concentrations used for stimulation.

(iii) 23 out of the 38 references are older than 5 years, and most of them are older than 10 years. While it is surely very important to refer to these older findings, the authors may include more knowledge from recent years. This may include references about centrosome positioning in immune cells and motile cells (e.g., PMID32379884, PMID29934494, PMID30944468, PMID37987147, PMID36398880, PMID38627564, PMID33634136) and the actin cytoskeleton at the centrosome (PMID33609453), PMID33184056, PMID36111670.

The references have been updated and more recent publications added.

(iv) The first paragraph of the discussion (lines 253 - 262) needs references.

Some references have been added to this paragraph.

(v) Some Figures maybe combined as the findings are closely related (e.g., Figures 2 and 3; and Figures 5 and 6).

The figures have been modified according to this suggestion.

(vi) Line 94, Supply. Fig. 1: the authors that the chemokine receptor CXCR4 has a uniform distribution in non-stimulated cells. This is not directly evident in the images as there are areas of more and less signal. It would be important to clearly describe this in the text. Further, the labelling of the figure would benefit from labelings such as 'cell 1', 'cell 2', etc to directly make clear that these are images from 3 representative cells.

We thank the reviewer for this remark. The CXCR4 distribution is indeed not strictly uniform. We now use the fire-colour coding which makes this point more obvious in images (Fig Supp 1).

However, although some high spots of CXCR4 accumulation can be observed, they are not associated to the location of centrosome and, can thus not explain the preferential position of the polarization axis we observe. This point is discussed in the new version of the manuscript (lines 113-120).

Furthermore, in the new figure concerning CXCR4 distribution (Fig Supp 1), we labelled each image with " cell x" in order to explicit that this figure displays different representative cells.

(vii) Different centrin isoforms exist (centrin 1, 2, 3). It should be mentioned in the results and methods section, which isoform was used for their genetic constructs (e.g., centrin-GFP, centrin-VCA).

We have corrected this omission. The fact that centrin1 isoform was used is now mentioned in the text, the legends and in the methods part.

Significance

This manuscript employs a lymphoblastic cell line called CEM T cells as a model for T lymphocytes. Using imaging of these cells on 2D substrates with and without chemokine, the authors identify a PKA-controlled actin pool at the centrosome that appears to regulate the local site of protrusion formation during cell polarisation. This is an interesting finding that adds to the knowledge of (i) the functions of centrosome positioning and (ii) the functions of the actin cytoskeleton at the centrosome. Thus, the study will be interesting to readers in the centrosome and migration fields. To broaden the scope of the manuscript, the findings could be tested in

primary T lymphocytes and mechanistically address the role of microtubules within the described process.

Reviewer #3

SUMMARY

Using a variety of live- and fixed-cell imaging techniques, the authors make correlative and causative connections between chemokine-stimulated increases in cAMP and localized PKA activity, positioning and F-actin content of the centrosome, and cell polarity in T-cells.

MAJOR COMMENTS

1. The authors state that "uniform CXCR4 labelling is observed in unstimulated cells" (ln94-95). While the panels in Supp Fig1a show a pattern in unstimulated cells that is obviously less dramatically asymmetrical than seen for stimulated (and, importantly, already polarized) cells in Supp Fig1b, the labelling in unstimulated cells is still far from uniform, as there is considerable heterogeneity of signal intensity along the perimeter. This is important, given that they are looking at a membrane receptor, even small fluctuations in which may be greatly amplified and thus have considerable effects on symmetry downstream. The authors should either quantify the intensity (e.g. signal as a function of polar coordinate value) or soften their language to more accurately reflect the data.

We thank the reviewer for this remark. The distribution of CXCR4 receptors is indeed not strictly uniform even in unstimulated cells. In order to make this point more obvious, we have used the fire-colour coding for CXCR4 distribution in Fig Supp1. As mentioned in the response to the reviewer #2, high spots of CXCR4 accumulation exist but they are not associated to the location of centrosome and thus, cannot explain the preferential position of the polarization axis we observe. This point is discussed in the new version of the manuscript (lines 113-120).

2. Regarding the analysis of polarization events in Fig. 1c and 1d, it is not clear how, exactly, the time point for the cortex opening is determined. For example, in the sample images in 1c, would the +110s or the +340s time point be used? The reason this is important is that the angle seems to change with time (at least in the example given) and there are also heterogeneities (specifically, decreases) in SiR-actin intensity along the cortex that precede cortex opening. Thus, it is not clear whether the cortex begins to open in closer proximity to the centrosome or whether the centrosome is further aligned after the cortex opens.

First of all, we would like to apologize that there is an error in the time labeling in Fig 1c. The third image corresponds to + 210s and not +110s. This has been corrected in the new Fig 1c. In this example, +210 s and not +340 s has been considered as the opening of the actin cortex.

More generally, the time for angle determination was performed by watching movies. Images were analyzed only relatively to previous and the next ones. Indeed, we first determined the period when 1/ SiRActin labeling stably decreases at one pole of the cell and 2/ the cell simultaneously deforms (transmitted light pictures). Playing the sequence backward, allowed us to determine the beginning of cortex opening. This time was then the reference for measuring the polarization

angles. Thus, transient SiRActin heterogeneities we could indeed observe sometimes, were not considered. These precisions have been added to the Methods section (lines 559-564)

3. Also regarding Figure 1d, it is not clear how many cells and experimental replicates are represented in the data - the Results text reports 60 cells from 11 experiments (ln106) but the legend reports 58 cells (ln564) without mention of experimental replicate number.

We thank the reviewer for pointing out this error which we have corrected. The correct number of cells is 60. These cells are from 11 different experiments. We pooled all individual values of the 60 cells to establish angle distribution.

Also, while the rose plot is useful, it is important to have statistical analysis on the skewness of the response and/or to report something other than the average angle - for example, the percentage of cells with a cortex opening in the same 90-degree quadrant as the centrosome. For more clarity, the position of the median has been added to the rose plot. Furthermore, a pie plot reports now the distribution of the angles (Fig 1d).

Finally, it might be clearer to the reader to have the rose plot and the model cell oriented in the same direction.

We agree to this suggestion and the cell (Fig 1c), the schematic drawing and the rose plot (Fig 1d) are all three oriented in the same direction.

4. For their PKA inhibition experiments, the authors introduce H89 as "a competitive inhibitor of ATP on the PKA catalytic subunit" (ln138-139). H89 is a very non-specific inhibitor, as demonstrated by Davies et al (PMID 10998351) and reviewed by Lochner & Moolman (PMID 17214602) and should be introduced more accurately.

H89, as a competitive inhibitor of ATP, is indeed not specific to PKA. At the low concentration we used, few other kinases can be inhibited. We now introduce more precisely the inhibitor and add references (lines 152-155 and 172-173).

To their credit, the authors use an orthogonal approach of PKA inhibition with a cAMP analog and see comparable effects. Those data, currently in Supp Fig. 3, should be included as primary data, given that the H89 data can only be correctly interpreted in the context of the Rp-cAMPS data (this applies to both Fig3 and Fig4).

As suggested, the data concerning Rp-8CPT-cAMPS are now included in the main figures (Fig 2g & Fig 3d) and the text has also been modified (lines 172-178 and 203-206).

5. As part of Fig.5, the authors state "AKAP450 is a type II AKAP i.e. it is able to bind RII subunits of PKA. In order to determine whether this AKAP allows a [sic] compartmentalization of the PKA activity responsible for centrosomal F-actin regulation, we used the specific peptide (Ht31)". Ht31 broadly inhibits PKA anchoring; its effect is not specific for any individual AKAP, including AKAP9/AKAP450.

We completely agree with this point and did not want to mean that Ht31 was specific to AKAP450. Therefore, the sentence has been modified to be more explicit (lines 234-236).

Moreover, the authors neither show/confirm PKA localization to the centrosome nor its displacement with the indicated concentration of Ht31, and they do not include data that PKA is displaced from the centrosome with any greater specificity or sensitivity than its displacement from any other subcellular location. Therefore, this statement (as well as the claim in the Abstract that "a specific pool of protein kinase A) (In15) by the authors is not accurate. The authors should, at the very least, re-word the statement and, for the sake of rigor and support of their hypotheses, confirm that PKA (subunits and/or activity) is displaced from the centrosome.

We thank the reviewer for raising this fair point. We have performed experiments to determine the distribution of PKA (immunofluorescence, using an antibody against PKA C α). Although the labeling is not very good, we can evidence a slight accumulation of the protein around the centrosome. This enrichment is statistically reduced in the presence of the inhibitory peptide Ht31. This suggest that 1) a pool of PKA accumulates around the centrosome and 2) it is displaced by Ht31.

These results as well as the corresponding images are presented in Figure Supp 7 and described in the new version of the manuscript (lines 236-245).

6. The experiments & results using VCA-centrin-GFP are very intriguing. However, it is crucial that primary data (i.e. photomicrographs/panels of fluorescent images of centrin/VCA-centrin localization, centrosomal F-actin, and roundness) be included for the readers' inspection. Also, it is not clear whether the graphically summarized data on centrosomal F-actin and roundness (Fig. 6b) represent analysis of cells before and after CXCL12 stimulation, or only before or only after. If either of the latter, analysis of these parameters both before and after stimulation should be included.

Representative images of cells expressing centrin/centrin-VCA are now provided and shown in Fig Supp 8. They show that centrin-VCA expressing cells are less deformed (a, b) but also display a higher quantity of F-actin around the centrosome (a).

The effect of stimulation in centrin-VCA expressing cells is now shown in Figure 4e. As mentioned in response to the reviewer #2 (iv), in cells expressing the centrin-VCA construct, the level of actin at the centrosome is higher than in control cells. However, upon chemokine stimulation, the signaling pathways leading to actin reduction are still active so that the level of centrosomal actin decreases (insignificantly different to the control) and the cells do not deform (insignificantly different to the control). These results are now discussed in the manuscript (lines 283-287).

MINOR COMMENTS

1. The writing is generally clear and accurate, but often somewhat 'choppy'. As one of many examples: "As depicted in Figure 1b, in this configuration, we observed that cell deformation starts rapidly after stimulation. The mean delay between cell stimulation and the time when cells start to deform is 112 {plus minus} 10 s (n=48 cells)" could be re-written as "As depicted in Figure 1b, in this configuration, we observed that cell deformation starts rapidly after stimulation, with a mean delay between cell stimulation and initiation of cell deformation of 112 {plus minus} 10 s (n=48 cells)." This is completely stylistic, of course, and would simply (albeit slightly) improve the readability of the work.

Full Revision

We have tried to improve the writing.

2. The authors comment that the contribution of calcium, PI3K, and cAMP signaling "in the early processes allowing the establishment of the asymmetric distribution of cellular components and of the polarity regulators is still elusive" (ln33-35) seems a bit overstated, as there have been numerous, impactful contributions investigating each of those pathways.

We agree that these actors have been already clearly shown to be involved in polarization and/or migration as reported in the review by V. Niggli. However, the precise role they play in the initiation of T cell polarization is not clear. The sentence concerning this point has been modified in the introduction (lines 36-40).

3. The work seems to start off as being focused on symmetry breaking rather than polarization, but this can be mitigated through rewriting the Introduction.

Symmetry breaking is for us the initial step (prerequisite) for cell polarization. Two sentences have been modified to clarify this point (lines 46-47 and 74-75).

4. The phrase "the major one" (ln47), presumably referring to one of "several local signaling poles" (earlier in ln47) is ambiguous and should be reworded (e.g. "the pole with the highest density of receptors").

The sentence has been modified (lines 53-54).

5. The phrase "variations of cAMP after CXCL12 addition upon dynamic cell imaging" (ln112) is not clear.

The sentence has been modified (lines 124-125).

6. The authors may want to reconsider the use of an ellipsis (ln25), which stands out as somewhat informal for a scientific manuscript.

The sentence has been reworded (lines 27-29).

7. There are several typographical errors throughout that should be addressed (e.g. "AMPc" rather than "cAMP" in the header of Fig7b.; "we were able establish" (ln 130); "while PKA are rapidly activated" (ln133)).

We have corrected typographical errors.

8. The figure legends most often read more like miniature, repeated results sections than detailed descriptions of experimental details and data processing, analysis and depiction.

Figure legends have been rewritten.

SIGNIFICANCE

Directional cell migration is a fundamentally important aspect of cell biology. Understanding the molecular mechanisms that govern cellular symmetry breaking, polarization, and migration are - in turn - important for a fuller understanding of how cells efficiently move from location to location. T cells, which are highly dependent on efficient and dynamically, cytokine-directed migration for

Full Revision

their physiologic function, are an excellent model system in which to unravel such molecular mechanisms. The authors efforts to connect localized cytokine-initiated signaling events with changes in centrosomal actin decoration and thence into cell polarity are, therefore, of considerable potential significance.

Referee #1:

The study visualises early events in T cell activation-driven symmetry breaking for cell migration. The authors propose that symmetry breaking is not random, but occurs near the centrosome and this is preceded by F actin depolarization, regulated by Ca^{++} signalling-induced accumulation of cAMP and local PKA activation. The finding is interesting, novel and sheds light on the very early stages of T cell activation, albeit in a 2D environment falling short in integrating cell cell interactions during early activation.

The key phenomenon is studied in CEM T ALL cells with some support from human primary T cells from peripheral blood. Overall, the conclusions are largely supported by the data, but limited to the in vitro setting and the use of a single chemokine.

There are some issues that need attention to increase general interest and possible impact.

1) Figure S2C is missing, and I think on several occasions, supplemental figures are not called out properly e.g S5, please check.

2) Also, resolution of suppl. images is very poor and I do not see centrosomal co-staining with human PBL in most panels

3) It is interesting to see that inhibition of PKA using H89 vs. Rp-CPT differs in the sense that the more specific inhibitor is less powerful and the stimulus strength needs to be reduced to see effects. Does this mean other kinases, maybe hit by H89 at the concentration used, are involved - could be discussed

4) If I read the model correctly, loss of centrosomal F actin is driven by PKA activity. This should free up space/facilitate MT nucleation, yet inhibition of actin removal has no impact on MT nucleation - how can this be reconciled with the proposed model?

5) I am missing some quality controls, especially related to the overexpression of Centrin-VCA fragments in primary T cells - how high is the transduction efficiency, also, is the symmetry breaking also near the centrosome in PBL T cells - e.g. in Fig. S3a -> misses quantification, as in Fig. 1d. Are CEM cells stably transduced with this expression construct?

6) migration assays are only done with CEM cells, not PBL T cells, which is a limitation.

Also, the authors often refer to the relevance of centrosomes in B cell migration and function, but primary mouse B cells lacking centrosomes are still competent to mount a humoral immune response, so, acentrosomal B lymphocytes appear to be able to find their way without centrosomal guidance of symmetry breaking. May be worth while discussing.

7) How is intracellular cAMP restricted/compartmentalized to activate only/mostly centrosomal PKA? Some discussion how such spatial control can be achieved would be helpful.

Referee #2:

The authors have addressed all of my comments in their revised manuscript.

The data presented appears to be solid and conclusive. Overall, the manuscript identifies that the localisation of the centrosome predicts the polarisation axis of T cells. Mechanistically, the authors show that a reduced amount of actin at the centrosome, regulated by the activity of PKA, is beneficial for cellular polarisation.

This is an interesting finding for the cell biology community as it provides another example of how centrosome positioning influences cell behaviour. However, the mechanistic basis of this finding - beyond the phenotypic observation and the involvement of PKA - remains largely elusive, as also indicated by the newly added data on microtubules. Therefore, the manuscript may be better suited for a more specialised cell biology journal within Review Commons, such as the Journal of Cell Science, or for a shorter report format like EMBO reports. Nonetheless, in this reviewer's opinion, the manuscript would be in principle ready for publication if representative example images of the microtubule are included.

Referee #3:

Directional cell migration is a fundamentally important aspect of cell biology. Understanding the molecular mechanisms that govern cellular symmetry breaking, polarization, and migration are - in turn - important for a fuller understanding of how cells efficiently move from location to location. T cells, which are highly dependent on efficient and dynamically, cytokine-directed migration for their physiologic function, are an

excellent model system in which to unravel such molecular mechanisms. Using a variety of live- and fixed-cell imaging techniques, the authors make correlative and causative connections between chemokine-stimulated increases in cAMP and localized PKA activity, positioning and F-actin content of the centrosome, and cell polarity in T-cells. In this revised version of the manuscript, the amount of respect, thought, and effort that the authors have put into responding to prior comments is outstanding and - more importantly - very effective in mitigating essentially all major and minor concerns, from this reviewer's perspective. The cumulative results - correlating the centrosome position with the site of symmetry breaking and polarization; showing that remodeling of actin at the centrosome facilitates polarization; and implicating centrosomal PKA as a regulator those actin dynamics - are, therefore, of considerable potential significance and are worthy of publication in the EMBO Journal.

Full Revision

Manuscript number: RC-2024-02542

Corresponding author(s): Clotilde, Randriamampita

1. General Statements [optional]

The two rounds of review have been added below (second round first) as well as the point by point reply.

In the manuscript, the revisions corresponding to the first round are in red, in green for the second one.

Second round of review

Referee #1:

The study visualises early events in T cell activation-driven symmetry breaking for cell migration. The authors propose that symmetry breaking is not random, but occurs near the centrosome and this is preceded by F actin depolarization, regulated by Ca⁺⁺ signalling-induced accumulation of cAMP and local PKA activation. The finding is interesting, novel and sheds light on the very early stages of T cell activation, albeit in a 2D environment falling short in integrating cell cell interactions during early activation.

The key phenomenon is studied in CEM T ALL cells with some support from human primary T cells from peripheral blood. Overall, the conclusions are largely supported by the data, but limited to the in vitro setting and the use of a single chemokine.

There are some issues that need attention to increase general interest and possible impact.

1) Figure S2C is missing, and I think on several occasions, supplemental figures are not called out properly e.g S5, please check.

We thank the reviewer for this remark. There were indeed several errors concerning figure citations in the text. Furthermore, we forgot to refer to the figure which displays microtubule distribution (Fig Supp 8). Therefore, several corrections (in green) have been performed in the revised version of the manuscript.

2) Also, resolution of suppl. images is very poor and I do not see centrosomal co-staining with human PBL in most panels

The Fig Supp 4a aims to exemplify PBT morphology in different conditions and therefore present images acquired at low magnification. The centrosomal actin can be better seen in Fig Supp 4c.

3) It is interesting to see that inhibition of PKA using H89 vs. Rp-CPT differs in the sense that the

more specific inhibitor is less powerful and the stimulus strength needs to be reduced to see effects. Does this mean other kinases, maybe hit by H89 at the concentration used, are involved - could be discussed

The fact that H89 can affect other kinase is mentioned in the manuscript (lines 152-155). We cannot formally exclude that a part of the observed effect is due to another H89-sensitive kinase. However, the fact that PKA is the kinase which regulates centrosomal actin is supported by the following considerations:

- Rp-8CPT cAMPS being a competitive inhibitor of PKA on the cAMP binding site, it is not surprising that it is less efficient when large amounts of cAMP are produced (strong chemokine stimulation). On the other hand, H89 is a competitive inhibitor on the ATP-binding site of the kinase. Its inhibitory potency is therefore less dependent on stimulation strength.
- The fact that H31 has a similar effect as H89, reinforces the hypothesis that cell deformation and centrosomal actin level are controlled by PKA.

To that respect, a few lines have been added in the discussion part (lines 338-345): "Our conclusions rely on temporal correlation between PKA activation and cell deformation / centrosomal actin reduction but also on pharmacological inhibitors targeting PKA (H89, Rp-8-CPT-cAMPS and Ht31). At H89 dose we used, H89 mainly inhibits PKA although we cannot rule out it also inhibits other kinases in our cells. However, H89 effect can be mimicked by two other agents specific to PKA: Ht31 and Rp-8-CPT-cAMPS. The fact that Rp-8-CPT-cAMPS is less efficient than H89 is probably linked to its mechanism of inhibition: it competes with endogenous cAMP so that its inhibition potency might be dependent on the quantity of cAMP produced, i.e. on the strength of chemokine stimulation."

4) If I read the model correctly, loss of centrosomal F actin is driven by PKA activity. This should free up space/facilitate MT nucleation, yet inhibition of actin removal has no impact on MT nucleation - how can this be reconciled with the proposed model?

Loss of centrosomal actin leads, as expected, to an increase of microtubule in the centrosomal area (Fig 4c). The surprising result is that PKA inhibition induces centrosomal actin increase but also of microtubules as well. These results are commented in the result part (256-264) and in the discussion (436-450).

5) I am missing some quality controls, especially related to the overexpression of Centrin-VCA fragments in primary T cells - how high is the transduction efficiency, also, is the symmetry breaking also near the centrosome in PBL T cells - e.g. in Fig. S3a -> misses quantification, as in Fig. 1d. Are CEM cells stably transduced with this expression construct?

The Centrin-VCA experiments have been performed in CEM T cells which were transiently transfected with the different constructs. Because of the GFP tag, only transfected cells have been taken into consideration for the analysis.

The first round of review mentions " If some key observations can be validated in primary T cells this would be perfect". We considered as key experiments the ones which show that in PBT 1) chemokine-induced PBT polarization is dependent on PKA activation and 2) the level of

centrosomal actin is under the control of this kinase in resting as well as in chemokine-stimulated PBT. These experiments are presented in Fig Supp4.

6) migration assays are only done with CEM cells, not PBL T cells, which is a limitation. Also, the authors often refer to the relevance of centrosomes in B cell migration and function, but primary mouse B cells lacking centrosomes are still competent to mount a humoral immune response, so, acentrosomal B lymphocytes appear to be able to find their way without centrosomal guidance of symmetry breaking. May be worth while discussing.

In our paper, we do refer to articles showing the importance of centrosome (and more specifically of centrosomal actin) in B cell polarization upon immune synapse formation (Obino et al, 2016), (Ibanez-Vega et al, 2019), (Bello-Gamboa et al, 2020). We are not aware of papers demonstrating the involvement of this process during B cell migration.

Furthermore, we have no evidence (and do not claim so) that T cells without centrosome do not polarize in response to chemokine. It is possible that in such cells, there would be no preferential axis of polarization or that some other parameters (for instance hot spots of chemokine receptors) would impose the position of this one. We unsuccessfully tried to generate T cells lacking centrosome (answer to reviewer #2, item iii in first round of review). We therefore do not know whether such cells would still present some intracellular accumulation of actin and whether this one, if present, would decrease upon chemokine stimulation thanks to global (i.e. uncompartimentalized) PKA activity.

Similar alternatives are possible in B lymphocytes lacking centrosomes for their motility and synapse formation. This could explain the normal immune response observed in such conditions (we guess that the referee refers to (Schapfl et al, Nature Communications, 2023)).

7) How is intracellular cAMP restricted/compartimentalized to activate only/mostly centrosomal PKA? Some discussion how such spatial control can be achieved would be helpful.

We do not claim that the cAMP signal is restricted in the cells. We propose on the other hand that PKA activity might be through control by AKAP450. This issue is addressed in the discussion section (361-367).

Referee #2:

The authors have addressed all of my comments in their revised manuscript.

The data presented appears to be solid and conclusive. Overall, the manuscript identifies that the localisation of the centrosome predicts the polarisation axis of T cells. Mechanistically, the authors show that a reduced amount of actin at the centrosome, regulated by the activity of PKA, is beneficial for cellular polarisation.

This is an interesting finding for the cell biology community as it provides another example of how centrosome positioning influences cell behaviour. However, the mechanistic basis of this finding - beyond the phenotypic observation and the involvement of PKA - remains largely elusive, as also indicated by the newly added data on microtubules. Therefore, the manuscript may be better suited for a more specialised cell biology journal within Review Commons, such as the Journal of

Cell Science, or for a shorter report format like EMBO reports. Nonetheless, in this reviewer's opinion, the manuscript would be in principle ready for publication if representative example images of the microtubule are included.

We apologize for this omission. Representative images displaying microtubule distributions in different conditions were provided in the revised version of the manuscript (supplementary figures) but we forgot to refer to the figure in the main text. This has now been corrected (lines 261): " As shown in Figure 4c **and in the example presented in Fig Supp 8**, while centrosomal actin decreases after CXCL12 stimulation, the level of polymerized microtubules increases."

Referee #3:

Directional cell migration is a fundamentally important aspect of cell biology. Understanding the molecular mechanisms that govern cellular symmetry breaking, polarization, and migration are - in turn - important for a fuller understanding of how cells efficiently move from location to location. T cells, which are highly dependent on efficient and dynamically, cytokine-directed migration for their physiologic function, are an excellent model system in which to unravel such molecular mechanisms. Using a variety of live- and fixed-cell imaging techniques, the authors make correlative and causative connections between chemokine-stimulated increases in cAMP and localized PKA activity, positioning and F-actin content of the centrosome, and cell polarity in T-cells. In this revised version of the manuscript, the amount of respect, thought, and effort that the authors have put into responding to prior comments is outstanding and - more importantly - very effective in mitigating essentially all major and minor concerns, from this reviewer's perspective. The cumulative results - correlating the centrosome position with the site of symmetry breaking and polarization; showing that remodeling of actin at the centrosome facilitates polarization; and implicating centrosomal PKA as a regulator those actin dynamics - are, therefore, of considerable potential significance and are worthy of publication in the EMBO Journal.

Dear Dr. Randriamampita,

Thank you for transferring your manuscript, which was previously peer-reviewed at Review Commons and the revised version was re-reviewed at The EMBO Journal by the original referees. I have now read your study and your response to the remaining concerns of the referees and would like to offer publication pending satisfactory minor revision as outlined below.

- Please provide source data as previously requested.
- Please submit a point-by-point response to the remaining referee concerns and implement the changes indicated in your email into the manuscript.
- As per our guidelines, please add a 'Data Availability Section', where datasets and computer code that were generated in the reported study should be listed in a structured manner and placed after the Methods section. If your study does not include datasets, please insert the following statement: This study includes no data deposited in external repositories (please see <https://www.embopress.org/page/journal/14693178/authorguide#dataavailability> for further information).
- Please submit the manuscript text in word format and remove the figures from the manuscript text files.
- Please provide the main and EV figures individually (as one production quality file per figure) without the legends in the separate files. The legends should remain in the manuscript text and be placed after the references section. The nomenclature of the Figure Supp 1, etc. should be corrected as Figure EV1, etc. Please see <https://www.embopress.org/page/journal/14693178/authorguide#figureformat> for further information.
- There are currently 9 Supp. Figures (which will be EV Figures). Please consider reducing the number of by combining some of them as they do not seem very busy.
- Please provide 3-5 keywords for your study. These will be visible in the html version of the paper and on PubMed and will help increase the discoverability of your work.
- Please rename the "Competing interests" section as "Disclosure And Competing Interests Statement".
- Please add the email address of the corresponding author on the title page.
- Please rename Materials & Methods as Methods.
- As per our format requirements, in the reference list, citations should be listed in alphabetical order and then chronologically, with the authors' surnames and initials inverted; where there are more than 10 authors on a paper, 10 will be listed, followed by 'et al.'. Please see <https://www.embopress.org/page/journal/14693178/authorguide#referencesformat>
- Please fill out and include an author checklist as listed in our online guidelines (<https://www.embopress.org/page/journal/14693178/authorguide>)
- Please include the funding information in the Acknowledgments section and remove the heading Funding section heading from the manuscript text.
- All research articles submitted as revised versions must include a structured methods section that includes a Reagents and Tools Table followed by a Methods and Protocols section. Please see <https://www.embopress.org/page/journal/14693178/authorguide#structuredmethods> for further information.
- Our production/data editors have asked you to clarify several points in the figure legends - Figure Legends (main + EV):
 - o Please indicate what */ **/ ***/ **** represents; if this represents p value(s), please indicate the statistical test used and where appropriate, specify the exact p value in the legend(s) of figure(s) 1A, 2C, F, G; 3A, C, D, E; 4B, C, E
- Papers published in EMBO Reports include a 'synopsis' and 'bullet points' to further enhance discoverability. Both are displayed on the html version of the paper and are freely accessible to all readers. The synopsis includes a short standfirst summarizing the study in 1 or 2 sentences (max 35 words) that summarize the paper and are provided by the authors and streamlined by the handling editor. I would therefore ask you to include your synopsis blurb and 3-5 bullet points listing the key experimental findings.
- In addition, please provide an image for the synopsis. This image should provide a rapid overview of the question addressed in the study but still needs to be kept fairly modest since the image size cannot exceed 550 (width) x 300-600 (height) pixels.

Thank you again for giving us to consider your manuscript for EMBO Reports, I look forward to your minor revision.

Kind regards,

Deniz Senyilmaz Tiebe

--

Deniz Senyilmaz Tiebe, PhD
Senior Scientific Editor
EMBO Reports

All editorial and formatting issues were resolved by the authors.

Dr. Clotilde Randriamampita
Université de Paris, Institut Cochin, INSERM, CNRS, F-75014 PARIS, France
Infection, Immunity and Inflammation
22 rue Méchain
Paris 75014
France

Dear Dr. Randriamampita,

Thank you for submitting your revised manuscript. I have now looked at everything and all is fine. Therefore, I am very pleased to accept your manuscript for publication in EMBO Reports.

Congratulations on a nice work!

Kind regards,

Deniz Senyilmaz Tiebe

--

Deniz Senyilmaz Tiebe, PhD
Senior Scientific Editor
EMBO Reports
